# FUNCTIONAL GEOMETRY GUIDED PROTEIN SEQUENCE AND BACKBONE STRUCTURE CO-DESIGN

## ABSTRACT

Proteins are macromolecules responsible for essential functions in almost all living organisms. Designing reasonable proteins with desired functions is crucial. A protein's sequence and structure are strongly correlated and they together determine its function. In this paper, we propose NAEPro, a model to jointly design Protein sequence and structure based on automatically detected functional and conserved sites. NAEPro is powered by an interleaving network of attention and equivariant layers, which can capture global correlation in a whole sequence and local influence from nearest amino acids in three dimensional (3D) space. Such an architecture facilitates effective yet economic message passing at two levels. We evaluate our model and several strong baselines on two protein datasets, $\beta$-lactamase and myoglobin. Experimental results show that our model achieves the highest binding affinity scores among the top-5, top-10 and top-30 candidates. These findings prove the capability of our model to design functional proteins. Furthermore, in-depth analysis further confirms our model's ability to generate highly effective proteins capable of binding to their target metallocofactors.

## 1 INTRODUCTION

Proteins are crucial macromolecules in almost all living organisms. A fundamental problem in protein engineering is designing novel proteins with specific biochemical functions such as catalytic activity (Park et al., 2006) and therapeutic efficacy (Sasportas et al., 2009). However, protein function arises from a complex coupling of sequence and structure: atoms that comprise protein structure must satisfy physical and chemical constraints while being "designable" in the sense that the amino-acid sequence should fold into that structure. This intricate connection between sequence and structure makes the task of functional protein design exceptionally challenging.

Recently, deep learning methods have witnessed impressive progress on different aspects of protein design (Ding et al., 2022). A representative class of work are pipeline-based approaches. These approaches typically start by prioritizing the design of the structure (Trippe et al., 2022; Yim et al., 2023; Lin & AlQuraishi, 2023; Watson et al., 2023), followed by the utilization of existing inverse folding models (Ingraham et al., 2019; Jing et al., 2020) like ProteinMPNN (Dauparas et al., 2022) to determine sequences that can fold into the specified structure. This kind of methods have been proven that they can generate new protein sequences with desired functions (Sumida et al., 2023; Zhou et al., 2023). Another line of methods is to co-design protein sequence and structure through cross-conditioningShi et al. (2022) . Although their approach is applicable to proteins of various topologies, their model relies on prespecified secondary structure to achieve general protein design, which cannot guarantee the designed proteins exhibit the desired functions. Wang et al. (2022) provide a solution to fill in additional sequence and structure given functional motifs. However, this approach assumes that biologists already possess knowledge of the motifs associated with the target proteins, thereby potentially restricting the model's applicability. How can we design functional and consistent protein sequence and structure efficiently?

In this paper, we propose NAEPro to jointly design protein sequence and structure guided by automatically detected meaningful protein fragments. We are motivated by the established wisdom that a protein's functionality is closely tied to its functionally critical sites, also known as motifs. Hence, by co-designing both elements based on these functional motifs, we can create functional proteins with consistent sequences and structures. Specifically, NAEPro is an interleaving network

consisting of stacked neighborhood attentive equivariant layers (NAELs). Each NAEL is composed of two integral components: a global attention sub-layer and a neighborhood equivariant sub-layer. The global attention sub-layer aims to capture global correlations across the entire protein sequence to discover favorable amino-acid combinations within one protein family. The neighborhood equivariant sub-layer is designed to filter out distant and potentially noisy information, concurrently gathering effective messages from the nearest amino acids in 3D space. This architectural design facilitates information exchange with varying levels of granularity, promoting a comprehensive interaction among different residues. This enhanced information exchange, in turn, contributes to more consistent and stably folded proteins. It is worth mentioning that NAEPro updates sequence and structure features of all residues in an one-shot manner, leading to a much more efficient design process.

We carry out experiments on two metalloproteins, including $\beta$-lactamase and myoglobin. The contribution of this paper are listed as follows:

- We propose NAEPro to jointly design protein sequences and backbone structures. This model is powered by the innovative neighborhood attentive equivariant layers (NAELs).
- Experiments show that NAEPro achieves highest binding affinity in all cases. Additionally, among the randomly selected 20 cases from top-100 candidates, $100\%$ and $90\%$ of them are highly potential to bind the corresponding metallocofactos, respectively for myoglobin and $\beta$-lactamase. NAEPro can even generate myoglobin sequences with amino acid identity rate as low as $66.0\%$ compared to the closest matches in Uniprot, yet they bear a remarkable resemblance to natural protein structures, with a RMSD of $0.458$Å.
- Our model is at least 17x faster than all the representative baselines. We will release our datasets, code and models.

## 2 RELATED WORK

**Protein Sequence Design** Protein sequence design has been studied with a wide variety of methods, including traditional directed evolution (Arnold, 1998; Dalby, 2011; Packer & Liu, 2015; Arnold, 2018) and machine learning methods (Belanger et al., 2019; Angermueller et al., 2019; Moss et al., 2020; Terayama et al., 2021). Following the success of deep generative models, there are some work focusing on protein sequence design with specific functions, aka. fitness. They either search satisfactory sequences using deep generative models (Brookes & Listgarten, 2018; Brookes et al., 2019; Madani et al., 2020; Kumar & Levine, 2020; Das et al., 2021; Hoffman et al., 2022; Melnyk et al., 2021; Anishchenko et al., 2021; Ren et al., 2022), or directly generate protein sequences applying deep generative models (Jain et al., 2022; Song & Li, 2023). Another class of methods focus on inverse-folding problem (Fleishman et al., 2011; Ingraham et al., 2019; Xiong et al., 2020; McPartlon et al., 2022; Hsu et al., 2022),which targets at producing a protein sequence that can fold into a given structure. Both approaches lack consideration for 3D structure design, resulting in constrained accuracy and novelty in the design outcomes.

**3D Protein Design** Wang et al. (2022) propose Inpainting to reconstruct both missing protein sequences and structures using provided motifs. However, it's essential to note that this approach necessitates the prior specification of motifs for the target proteins, which requires domain-specific expertise. Another class of methods focus on novel protein structure design (Trippe et al., 2022; Yim et al., 2023; Lin & AlQuraishi, 2023; Watson et al., 2023) and then apply an inverse folding model to identify a sequence based on the given backbone structure, which may not fully utilize the mutual constraints between protein sequence and structure. Anand & Achim (2022) first propose to co-design protein sequence and structure conditioning on given secondary structures (SS). Following their work, Shi et al. (2022) propose to realize general protein design conditioning on SS and binary contact map. However, designing protein relying on its topology cannot guarantee the designed proteins have the desired functions.

## 3 PROPOSED METHOD: NAEPRO

A protein consists of a chain of amino acids (also called residues) connected by peptide bonds, which folds into a proper 3D structure. Let $\mathcal{A}$ be the set of 20 common amino acids. We denote the sequence of a $N$-residue protein by $\boldsymbol{s} = \{s_1, s_2, ..., s_N\} \in \mathcal{A}^N$ and their $C_\alpha$ coordinates by $\boldsymbol{x} = [\boldsymbol{x}_1, \boldsymbol{x}_2, ..., \boldsymbol{x}_N]^T \in \mathbb{R}^{N \times 3}$. For residue type $s_i \in \mathcal{A}$, where $i \in \{1, 2, ..., N\}$, we denote its

one-hot encoding as $\boldsymbol{s}_i = \text{onehot}(s_i)$. We denote the scaffold, indexed as $\mathcal{F} = \{1, 2, ..., N\}$, within which the meaningful fragment index set is represented by $\mathcal{M}$.

The problem studied in this paper can be formulated as follows: given a meaningful fragment index set $\mathcal{M}$, its residue set $\boldsymbol{s}_{\mathcal{M}}$ and its corresponding 3D coordinates $\boldsymbol{x}_{\mathcal{M}}$, generate a protein sequence and $C_\alpha$ coordinates of $N$ residues. Essentially, we aim to learn a generative model with probability $P(\boldsymbol{s}, \boldsymbol{x}|\boldsymbol{s}_{\mathcal{M}}, \boldsymbol{x}_{\mathcal{M}})$ where $\boldsymbol{s}$ is a $N$-residue protein sequence and $\boldsymbol{x}$ are their $C_\alpha$ coordinates. The task is challenging because the designed protein sequence $\boldsymbol{s}$ needs to fold into the structure $\boldsymbol{x}$. Meaningful fragments here represent protein functional and conserved sites, in which functional sites vary from setting to setting. For example, they are active sites for *de novo enzyme design* (Richter et al., 2011), while they are binding sites for *de novo binder design* (Gainza et al., 2023).

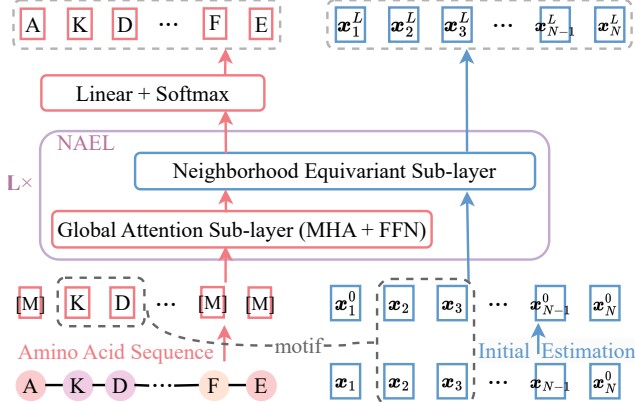

Figure 1: NAEPro architecture, which consists of $L$ stacked neighborhood attentive equivariant layers (NAELs). Each NAEL is composed of one global attention sub-layer and one neighborhood equivariant sub-layer.

## 3.1 OVERALL MODEL ARCHITECTURE

We propose a model named NAEPro to simultaneously predict the protein sequence $\boldsymbol{s}$ and its 3D backbone structure $\boldsymbol{x}$ given fragment residues $\boldsymbol{s}_{\mathcal{M}}$ and their $C_\alpha$ coordinates $\boldsymbol{x}_{\mathcal{M}}$. NAEPro is a deep neural network consisting of $L$ stacked neighborhood attentive equivariant layers (NAELs), which is depicted in Figure 1. It takes fragment residues and the corresponding coordinates as the input and outputs probabilities of amino acid type for all residues and coordinates of a protein. Each NAEL is composed of a global attention sub-layer using Transformer (Vaswani et al., 2017) and a neighborhood equivariant sub-layer to incorporate information from nearby residues based on $C_\alpha$ coordinates. Our design rationale centers on computing contextualized representation for each residue while simultaneously enhancing the estimation of each $C_\alpha$ coordinate. These two are strongly correlated. Ideally, each residue should receive contextual information from all residues within the protein sequence, as well as from its closest neighbors determined by the 3D structure of the protein.

Suppose $\theta$ are NAEPro parameters. We can formulate the joint sequence and structure probability as:

$$P(\boldsymbol{s}, \boldsymbol{x}|\boldsymbol{s}_{\mathcal{M}}, \boldsymbol{x}_{\mathcal{M}}; \theta) = P(\boldsymbol{s}|\boldsymbol{s}_{\mathcal{M}}, \boldsymbol{x}_{\mathcal{M}}; \theta) \cdot P(\boldsymbol{x}|\boldsymbol{s}_{\mathcal{M}}, \boldsymbol{x}_{\mathcal{M}}; \theta)$$
$$P(\boldsymbol{s}|\boldsymbol{s}_{\mathcal{M}}, \boldsymbol{x}_{\mathcal{M}}; \theta) = \Pi_{i=1\&i\notin\mathcal{M}}^{N} P(\boldsymbol{s}_i|\boldsymbol{s}_{\mathcal{M}}, \boldsymbol{x}_{\mathcal{M}}; \theta), \quad P(\boldsymbol{x}|\boldsymbol{s}_{\mathcal{M}}, \boldsymbol{x}_{\mathcal{M}}; \theta) = \Pi_{i=1\&i\notin\mathcal{M}}^{N} P(\boldsymbol{x}_i|\boldsymbol{s}_{\mathcal{M}}, \boldsymbol{x}_{\mathcal{M}}; \theta)$$
$$P(\boldsymbol{s}_i|\boldsymbol{s}_{\mathcal{M}}, \boldsymbol{x}_{\mathcal{M}}; \theta) = \text{Softmax}(W_{\mathcal{A}} \cdot \boldsymbol{h}_i^L), \quad \boldsymbol{x}_i|\boldsymbol{s}_{\mathcal{M}}, \boldsymbol{x}_{\mathcal{M}} \sim \mathcal{N}(\boldsymbol{x}_i^L; \lambda I)$$
$$(1)$$

where $W_{\mathcal{A}}$ is the embedding matrix for 20 naturally amino acids, $\boldsymbol{h}_i^L$ is the output embedding for $i^{th}$ residue at the last layer of our network and $\boldsymbol{x}_i^L$ is the corresponding output coordinate. $\mathcal{N}(\boldsymbol{x}_i^L; \lambda I)$ is the Gaussian distribution with mean $\boldsymbol{x}_i^L$ and covariance matrix $\lambda I$ ($I$ is the identity matrix). $\lambda$ is a hyperparameter. To find the optimal $\theta$, we maximize the conditional log likelihood (or equivalent to minimizing the negative log likelihood):

$$\theta^* = \arg\min_{\theta} \mathcal{L}(\theta) = \arg\min_{\theta} -\log P(\boldsymbol{s}, \boldsymbol{x}|\boldsymbol{s}_{\mathcal{M}}, \boldsymbol{x}_{\mathcal{M}}; \theta)$$
$$= \arg\min_{\theta} -\sum_{i=1\&i\notin\mathcal{M}}^{N} \log P(\boldsymbol{s}_i|\boldsymbol{s}_{\mathcal{M}}, \boldsymbol{x}_{\mathcal{M}}; \theta) - \sum_{i=1\&i\notin\mathcal{M}}^{N} \log P(\boldsymbol{x}_i|\boldsymbol{s}_{\mathcal{M}}, \boldsymbol{x}_{\mathcal{M}}; \theta)$$
$$(2)$$

For simplicity, we omit the number of protein samples in a dataset. The second log likelihood function can be further simplified as:

$$\log P(\boldsymbol{x}_i|\boldsymbol{s}_{\mathcal{M}}, \boldsymbol{x}_{\mathcal{M}}; \theta) = \log\{\frac{1}{\sqrt{(2\pi)^3}} \exp\left(-\frac{1}{2}(\boldsymbol{x}_i - \boldsymbol{x}_i^L)^T \lambda(\boldsymbol{x}_i - \boldsymbol{x}_i^L)\right)\} = -\frac{\lambda}{2}||\boldsymbol{x}_i - \boldsymbol{x}_i^L||_2^2 + \text{const} \quad (3)$$

Therefore, the overall training objective is:

$$\mathcal{L}(\theta) = -\sum_{i=1 \& i \notin \mathcal{M}}^{N} \log P(\boldsymbol{s}_i | \boldsymbol{s}_{\mathcal{M}}, \boldsymbol{x}_{\mathcal{M}}; \theta) + \frac{\lambda}{2} \sum_{i=1 \& i \notin \mathcal{M}}^{N} ||\boldsymbol{x}_i - \boldsymbol{x}_i^L||_2^2 \tag{4}$$

To calculate the prediction probability, we propose the neighborhood attentive equivariant layer (NAEL) to infer non-fragment residue representations and their 3D coordinates based on given meaningful fragment residues and the corresponding positions. Our key insight is to facilitate thorough information exchange among residues at two levels, global sequence level and local Euclidean distance-based neighborhood.

## 3.2 NAEL GLOBAL ATTENTION SUB-LAYER

This sub-layer computes global contextual embeddings for all residues. This sub-layer does not consider the closeness of residues in 3D space. Considering that homologous sequences encode rich semantic features among constituent residues, we allow every residue to attend to all other residues across the whole protein sequence to facilitate information flow through the overall sequence.

We adopt the Transformer layer (Vaswani et al., 2017) to compute global contextual embeddings. Specifically, each transformer layer is composed of one multi-head self-attention sub-layer (MHA) and one fully connected feed-forward network (FFN). A residue connection and a layer normalization are employed after each of the two sub-layers. Thus, the calculation of global sequence attention can be formulated as follows:

$$\boldsymbol{h}_i^{l+0.5} = \text{LayerNorm}\left(\text{FFN}(\tilde{\boldsymbol{h}}_i^{l+0.5}) + \tilde{\boldsymbol{h}}_i^{l+0.5}\right), \quad \tilde{\boldsymbol{h}}_i^{l+0.5} = \text{LayerNorm}\left(\text{MHA}(\boldsymbol{h}_i^l, \boldsymbol{H}^l) + \boldsymbol{h}_i^l\right) \tag{5}$$

where $\boldsymbol{h}_i^l$ is the $i^{th}$ residue representation at $l^{\text{th}}$ layer and $\boldsymbol{H}^l = [\boldsymbol{h}_1^l, \boldsymbol{h}_2^l, ..., \boldsymbol{h}_N^l]^T$. The input residue embeddings for the first layer are either taken from an embedding lookup table for fragment residues, or initialized with a special [mask] token embedding for non-fragment residues:

$$\boldsymbol{h}_i^0 = \begin{cases} \boldsymbol{s}_i^T W_{\mathcal{A}}, & i \in \mathcal{M}, \text{ i.e. } i^{th} \text{ residue is in fragment,} \\ \text{Emb([mask])}, & \text{otherwise} \end{cases} \tag{6}$$

where $W_{\mathcal{A}} \in \mathbb{R}^{20 \times d}$ is the embedding matrix for 20 amino acids and $d$ is the dimensionality of residue embeddings.

## 3.3 NAEL NEIGHBORHOOD EQUIVARIANT SUB-LAYER

On a protein, a residue's representation and 3D position can be influenced by its nearest neighboring residues. We intend to model such impact with a properly designed subnetwork while keeping the equivariance under 3D translation, rotation and reflection. To this end, we propose the neighborhood equivariant sub-layer. This sub-layer includes three components: neighborhood message update, coordinate update and residue update (Figure 2). It

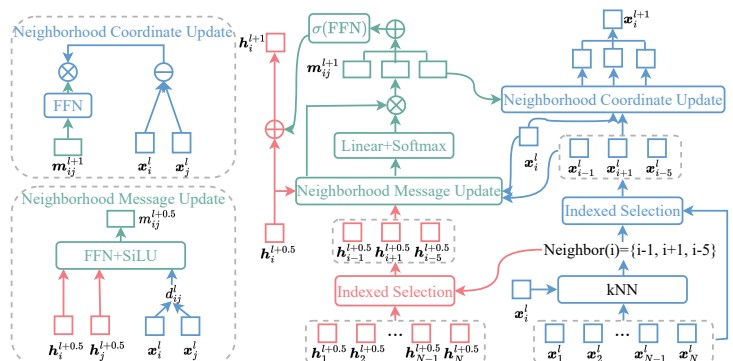

Figure 2: Neighborhood equivariant sub-layer: neighborhood message update (green), coordinate update (blue) and residue update (red). Indexed selection is choosing $\boldsymbol{x}_j$ (or $\boldsymbol{h}_j$) where $j^{th}$ residue is in the k-nearest neighbors (kNN) of $i^{th}$ residue.

will calculate the updated 3D coordinates of residues based on impact from their neighbors and the updated residue embeddings considering local neighbors. Specifically, we calculate k-nearest neighbors for each residue based on the Euclidean distances of residue coordinates. Updating residue representations and coordinates in 3D space with only nearest neighbors enables more efficient and

economic message passing compared to prior approaches which compute messages on the complete pairwise residue graph.

**Neighborhood message update.** We first compute distances among residues using $C_\alpha$ coordinates, and select k nearest residues (Figure 2). We compute the messages between $i^{th}$ residue and its k-nearest neighbors (denoted as Neighbor(i)) as follows:

$$\boldsymbol{m}_{ij}^{l+0.5} = \text{SiLU}(\text{FFN}(\text{Concat}(\boldsymbol{h}_i^{l+0.5}, \boldsymbol{h}_j^{l+0.5}, ||\boldsymbol{x}_i^l - \boldsymbol{x}_j^l||_2)))$$

$$w_{ij}^{l+0.5} = \frac{\exp(W_a^l \boldsymbol{m}_{ij}^{l+0.5} + b_a^l)}{\sum_{j' \in \text{Neighbor(i)}} \exp(W_a^l \boldsymbol{m}_{ij'}^{l+0.5} + b_a^l)} \tag{7}$$

$$\boldsymbol{m}_{ij}^{l+1} = w_{ij}^{l+0.5} * \boldsymbol{m}_{ij}^{l+0.5}$$

where FFN is a two-layer fully connected feed-forward network with SiLU activation function after its first layer. Concat is concatenation operator and $||\boldsymbol{x}_i^l - \boldsymbol{x}_j^l||_2$ is the Euclidean distance of $i^{th}$ and $j^{th}$ residue coordinates at $l^{th}$ layer. $W_a^l \in \mathbb{R}^{1 \times d}$ and $b \in \mathbb{R}$ are the trainable attention parameters of $l^{th}$ layer.

**Neighborhood coordinate update.** We update the $C_\alpha$ coordinate of $i^{th}$ residue as a k-neasrest neighbor vector field in a radial direction (Figure 2). The $C_\alpha$ coordinate $\boldsymbol{x}_i^l$ at $l^{th}$ layer of $i^{th}$ residue is updated with the weighted sum of all relative differences $(\boldsymbol{x}_i^l - \boldsymbol{x}_j^l)_{\forall j \in \text{Neighbor(i)}}$:

$$\boldsymbol{x}_i^{l+1} = \boldsymbol{x}_i^l + \sum_{j \in \text{Neighbor(i)}} (\boldsymbol{x}_i^l - \boldsymbol{x}_j^l) \cdot \text{FFN}(\boldsymbol{m}_{ij}^{l+1}) \tag{8}$$

where FFN is again a two-layer fully connected feed-forward network with SiLU activation function after its first layer.

The input $C_\alpha$ coordinates for the whole model are either the given coordinates for fragment residues, or randomly initialized as 3D points on the spherical surface centered at its left residue since the observed Euclidean distances of every neighboring $C_\alpha$ pair are almost the same (around $r = 3.75$Å):

$$\boldsymbol{x}_i^0 = \begin{cases} \boldsymbol{x}_i, & i \in \mathcal{M}, \text{i.e. } i^{th} \text{ residue is in fragment,} \\ \boldsymbol{x}_{i-1}^0 + r \cdot [\sin \omega_1 \cos \omega_2, \sin \omega_1 \sin \omega_2, \cos \omega_1]^T, & \text{otherwise} \end{cases} \tag{9}$$

where $\omega_1$ and $\omega_2$ are sampled from polar coordinate system. $\omega_1$ is the angle to Z-axis sampled from Uniform$(0, \pi)$ and $\omega_2$ is the angle to X-axis sampled from Uniform$(0, 2\pi)$.

**Neighborhood residue update.** We update the $i^{th}$ residue representation by gathering information from its k-nearest neighbors through a gating mechanism (Figure 2):

$$\boldsymbol{c}_i^{l+1} = \sum_{j \in \text{Neighbor(i)}} \boldsymbol{m}_{ij}^{l+1}$$

$$\boldsymbol{h}_i^{l+1} = \boldsymbol{h}_i^{l+0.5} + \sigma(\text{FFN}(\boldsymbol{c}_i^{l+1})) \odot \boldsymbol{c}_i^{l+1} \tag{10}$$

where FFN is a two-layer fully connected feed-forward network with ReLU activation function after its first layer. $\sigma$ denotes the sigmoid activation function.

We stack $L$ layers of NAEL to form the whole model (Figure 1). The output embedding $\boldsymbol{h}_i^L$ and coordinate $\boldsymbol{x}_i^L$ for the $i^{th}$ residue are both from the last layer. Then the output probability of amino acid type for $i^{th}$ residue is calculated as:

$$P(s_i = a | \boldsymbol{s}_\mathcal{M}, \boldsymbol{x}_\mathcal{M}) = \frac{\exp(h_{i,a}^o)}{\sum_{a'=1}^{20} \exp(h_{i,a'}^o)}, \quad \boldsymbol{h}_i^o = W_\mathcal{A} \cdot \boldsymbol{h}_i^L \tag{11}$$

## 3.4 MEANINGFUL PROTEIN FRAGMENT MINING

The functionality of a protein is commonly linked to its functional motifs. Consequently, leveraging these functional motifs, we can design proteins with specific functions. In this context, we present a heuristic method for efficient extraction of protein functional and conserved sites. Given that proteins belonging to the same family often share similar functions, they are highly potential to have identical motifs. Therefore, we leverage multiple sequence alignments (MSAs) to mine meaningful protein fragments, including functional and conserved sites, within the same protein family. Specifically, we employ the ClustalW2 method (Anderson et al., 2011) to perform MSAs for each protein family. Subsequently, we designate residue sites as meaningful fragments when they surpass a prespecified $\tau\%$ identity threshold, as determined by the MSA results.

## 3.5 ANALYSIS ON $SE(3)$ EQUIVARIANCE

Equivariance plays a critical role in ensuring consistent and predictable performance in the nuisance transformations applied to the input data. Thus, we analyze the SE(3) equivariance of our model.

**Theorem 3.1.** *Let R denotes a rotation matrix from SO(3) group and $\boldsymbol{t} \in \mathbb{R}^3$ from the translation group. Our NAEL is $SE(3)$-equivariant: $\boldsymbol{H}^{l+1}, R\boldsymbol{x}^{l+1} + \boldsymbol{t} = \text{NAEL}(\boldsymbol{H}^l, R\boldsymbol{x}^l + \boldsymbol{t})$.*

**Corollary 3.2.** *Parameterizing NAEPro as a composition of L NAELs, and taking $l^{th}$ layer output $\boldsymbol{x}^{l+1}$ and $\boldsymbol{H}^{l+1}$ as the $(l + 1)^{th}$ layer input. NAEPro is SE(3)-equivariant.*

We provide a formal proof in Appendix A.

## 4 EXPERIMENTS

We conduct extensive experiments to examine the effectiveness of our proposed method. We aim to address the following questions:

- **Function (Q1)**: Do the NAEPro designed proteins exhibit the desired functions?
- **Efficiency (Q2)**: What is the inference computational complexity?

### 4.1 DATASETS

We evaluate our NAEPro on two metalloproteins: $\beta$-lactamase and myoglobin. $\beta$-lactamase binds zinc ion and myoglobin binds heme. These two belong to a large category of proteins, metalloproteins, which are proteins containing a metal ion cofactor. Metalloproteins play vital roles in a variety of cell functions, such as storage and transport of proteins. We thoroughly discuss the significance of these two metalloproteins in Appendix B.1. To obtain the data, we first extract all proteins in PDB belonging to the two proteins. Then we extract chain A for $\beta$-lactamase and all chains for myoglobin. We only keep proteins capable of binding metallofactors. Next we perform length filtering for both proteins, i.e., reserving $\beta$-lactamase longer than 200 and myoglobin longer than 100 to make sure the data are reasonable. Then we run MSAs to mine fragments. Finally, we randomly split each dataset into training/validation/test sets with the ratio $8 : 1 : 1$. The data statistics are given in Appendix B.2.

### 4.2 EXPERIMENTAL DETAILS

**Implementation Details** We use 6 NAELs in NAEPro. The global attention sub-layer parameters are initialized with released ESM-2 (Lin et al., 2022) parameters (esm2_t6_8M_UR50D). The hyperparameter $\lambda/2$ and $k$ are respectively set to 1.0 and 30. The mini-batch size and learning rate are set to 8 and 5e-4 respectively. The model is trained for 100 epochs with 1 NVIDIA RTX A6000 GPU card. The sequences are decoded using greedy decoding strategy. To make training easier, we employ an annealing training strategy for the first 10 epochs, which randomly sample (85% * (10 - epoch) / epoch) residues as pseudo fragments and use real fragments after 10 epochs. MSA threshold $\tau$ is respectively set to 30 and 18 respectively for myoglobin and $\beta$-lactamase as $\beta$-lactamase has three sub-classes and the sequence is highly variable with length ranging from 200 to over $1,000$.

**Baseline Models** We compare the proposed NAEPro against the following representative baselines: (1) **Hallucination** (Anishchenko et al., 2021) uses MCMC (Andrieu et al., 2003) incorporating a motif constraint into the acceptance score calculation. (2) **Inpainting** (Wang et al., 2022) recovers both sequence and structure based on the given protein segments. (3) **SMCDiff** (Trippe et al., 2022)+**ProteinMPNN** (Dauparas et al., 2022): We first apply SMCDiff to design a protein structure based on given motifs and use ProteinMPNN to generate a sequence based on the given structure. (4) **PROTSEED** (Shi et al., 2022) co-designs protein sequence and backbone structure based on secondary structure and binary contact map for general protein design. (5) **FrameDiff** (Yim et al., 2023)+**ProteinMPNN**: Similar to (3) but with a SE(3) invariant diffusion model to design structure. (6) **RFDiffusion** (Watson et al., 2023)+**ProteinMPNN**: Similar to (3) but with a different structure design model which is finetuned from the pretrained RoseTTAFold model (Baek et al., 2021).

To better analyze the influence of different components in our model, we also conduct the following ablation tests: (7) **NAEPro-w/o-gate** replaces the gating mechanism in residue updating process with a MLP like in previous graph neural network. (8) **NAEPro-w/o-kNN** replaces the k-nearest

| | Models | Top-5 ($\downarrow$) | Top-10 ($\downarrow$) | Top-30 ($\downarrow$) | Median ($\downarrow$) |
|---|---|---|---|---|---|
| $\beta$-lactamase | Hallucination | -6.98±0.01 | -6.87±0.02 | -6.69±0.05 | -6.29 |
| | Inpainting | -9.89±0.03 | -9.54±0.16 | -9.13±0.43 | -7.24 |
| | SMCDiff+ProteinMPNN | -9.10±0.01 | -9.05 ±0.02 | -8.98±0.01 | -6.97 |
| | PROTSEED | -9.88±0.21 | -9.51±0.41 | -9.01±0.62 | -7.31 |
| | FrameDiff+ProteinMPNN | -9.54±0.03 | -9.56±0.23 | -8.89±0.35 | -7.03 |
| | RFDiffusion+ProteinMPNN | -9.87±0.05 | -9.56±0.23 | -9.12±0.53 | -7.51 |
| | NAEPro | **-10.06±0.05** | **-9.79±0.10** | **-9.39± 0.12** | **-7.66** |
| myoglobin | Hallucination | -8.18± 0.01 | -8.07±0.03 | -7.97±0.23 | -7.25 |
| | Inpainting | -13.47±0.02 | -13.12±0.12 | -12.31±0.54 | -9.56 |
| | SMCDiff+ProteinMPNN | -11.37±0.03 | -11.12±0.31 | -10.87±0.42 | -8.76 |
| | PROTSEED | -13.21±0.13 | -12.89±0.42 | -11.98±0.52 | -10.23 |
| | FrameDiff+ProteinMPNN | -13.13±0.05 | -12.92±0.16 | -12.21±0.23 | -10.08 |
| | RFDiffusion+ProteinMPNN | -13.68±0.02 | -13.03±0.21 | -12.56±0.43 | -10.15 |
| | NAEPro | **-14.12±0.01** | **-13.85±0.10** | **-13.06±0.38** | **-10.74** |

Table 1: Model performance on two metalloprotein datasets. The unit system for top-5, top-10, top-30 mean and median binding affinity score is kcal/mol.

neighbor graph with complete pairwise residue graph. (9) **NAEPro-w/o-MFFN** replaces FFN with the self-attention mechanism in Transformer during message update. (10) **NAEPro-w/o-MCFFN** replaces FFN with the self-attention mechanism in message update and uses the attention weight to replace FFN in coordinate update. (11) **NAEPro-w-RandomInit** removes ESM2 parameter initialization. (12) EGNN+ESM2 concatenates EGNN and ESM2, and then is fientuned on the corresponding dataset.

**Evaluation Metrics** We calculate the binding affinity scores applying Gnina (McNutt et al., 2021) to evaluate the function of the designed proteins. Gnina is a docking tool which can assess how well the designed protein can bind to the corresponding metallocofactors. The lower the docking score, the better the binding affinity. We provide top-5, top-10, top-30 mean binding affinity score as well as median binding affinity score.

## 4.3 MAIN RESULTS

**Q1: NAEPro can generate proteins that have highest binding affinity.** In particular, Table 1 shows that our model achieves the lowest docking scores in all cases, which means our NAEPro can better bind the corresponding metallocofactors. Among all the baselines, RFDiffusion+ProteinMPNN is the current SOTA model and has been validated that it can generate proteins with desired functions. Table 1 shows that our model achieves even higher binding affinity than this model. It demonstrates that our designed proteins are highly potential to exhibit the desired functions.

**Q2: NAEPro inference is much faster.** We visualize the average design time on the test set of all models in Figure 7 (a). It shows our model is much more efficient than all other competitors. Inpainting and PROTSEED achieve the shortest inference time among all the baselines, but our model still outpaces them, being at least 17× faster (0.17s compared to 3.00s on $\beta$-lactamase). The reason is the global attention sub-layer and neighborhood equivariant sub-layer update features for all residues in an one-shot manner, leading to a much more efficient inference process. The detailed design time of all models are provided in Appendix C.1.

## 4.4 ABLATION STUDY: CONTRIBUTION OF MODEL COMPONENTS

Table 2 shows the results of ablation study on myoglobin. The most substantial degradation in performance occurs when applying complete pairwise residue graph (NAEPro-w/o-kNN) instead of k-nearest neighbors. This outcome could be attributed to the excessively long protein sequences, where distant residues in 3D space may not have meaningful connections. Forcing these distant residues to exchange information might introduce noise into the model. Replacing FFN with the self-attention mechanism during message update will also cause performance degradation (NAEPro-w/o-MFFN), while the additional removal of FFN in coordinate update (w/o-MCFFN) does not lead to a significant performance difference. It demonstrates that FFN in message update can better

| Models | Top-5 (↓) | Top-10 (↓) | Top-30 (↓) | Median (↓) |
|---|---|---|---|---|
| NAEPro | **-14.12±0.01** | **-13.85±0.10** | **-13.06±0.38** | **-10.74** |
| – w/o-gate | -12.74±0.40 | -12.50±0.37 | -12.07±0.41 | -9.63 |
| – w/o-kNN | -11.97±0.10 | -11.80±0.19 | -11.61±0.17 | -7.15 |
| – w/o-MFFN | -12.14±0.31 | -11.88±0.35 | -11.64±0.26 | -9.26 |
| – w/o-MCFFN | -12.26±0.36 | -11.99±0.37 | -11.69±0.30 | -9.22 |
| – w-RandomInit | -13.52±0.30 | -12.90±0.66 | -12.11±0.68 | -9.09 |
| EGNN+ESM2 | -13.30±0.61 | -12.77±0.69 | -12.07±0.65 | -9.63 |

Table 2: NAEPro and variants on myoglobin design. Notice that computing message only on k-nearest neighbors rather than on the full residues contributes to significant improvement in NAEPro. The unit system for top-5, top-10, top-30 mean and median binding affinity score is kcal/mol.

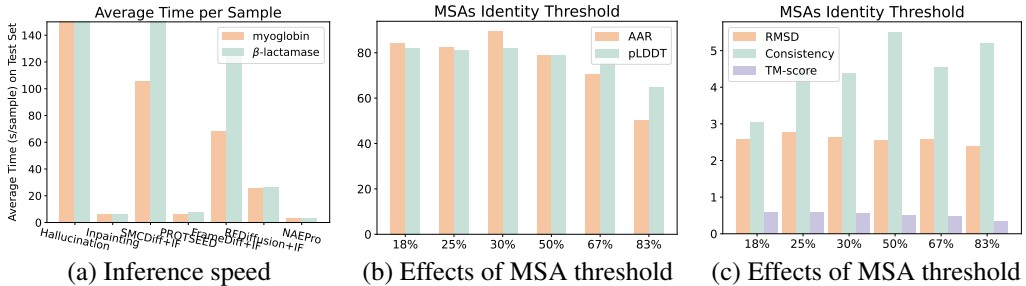

(a) Inference speed     (b) Effects of MSA threshold     (c) Effects of MSA threshold

Figure 3: Visualization of (a) inference speed of all models evaluated by average design time on test set. (b) and (c) model performance on myoglobin under different MSA identity thresholds.

gather information by considering both semantic and structural similarities. Leveraging the gating mechanism will slightly boost the model performance, and utilizing ESM2 initialization can further promote functional protein design.

## 5 ANALYSIS OF PROTEINS DESIGNED BY NAEPRO

### 5.1 EFFECTS OF MSA IDENTITY THRESHOLD

To gain insights into how the given meaningful fragments would influence the design quality, we provide the results on myoglobin under different MSA thresholds. Figure 7 (b) and (c) show a progressive decline in the quality of the designed sequences and structures as the MSA threshold increases. However, when the threshold remains below 30%, there are no substantial differences observed. To ensure high design quality while providing minimum information, we set the threshold to 30% for myoglobin. This choice aims to strike a balance between generating plausible proteins and fostering novelty.

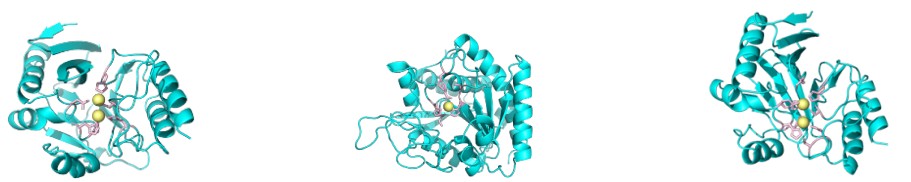

(a) B1 metallo-β-lactamase     (b) B2 metallo-β-lactamase     (c) B3 metallo-β-lactamase

Figure 4: Designed β-lactamases belonging to different subclasses: (a) B1, (b) B2 and (c) B3 metallo-β-lactamases.

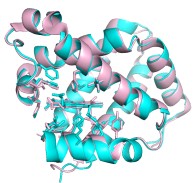
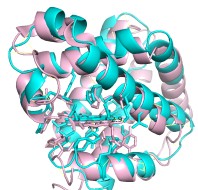
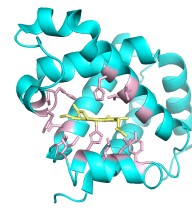

(a) binding at His-92          (b) Binding at His-89          (c) myoglobin with PDB id=1SPG

Figure 5: Designed myoglobins binding heme ligand at (a) 92-Histidine and (b) 89-Histidine, and respectively have a low amino acid identity rate of 66.0% and 26.7% to the most similar one in Uniprot ((c) PDB id=1SPG) but also with a low RMSD distance of (a) 0.458Å and (b) 3.943Å.

## 5.2 CAN NAEPRO DESIGNED PROTEINS BIND THE CORRESPONDING METALLOCOFACTORS?

To further test if our designed proteins are potential to have the desired functions, we evaluate the basic properties of metalloproteins, i.e. if they can bind their corresponding metallocofactors. Metalloproteins are required to contain metal ion cofactors to carry out their functions, and thus assessing the metallocofactor binding property is meaningful and important. Specifically, we first randomly select 20 cases from the top-100 sequences according to their pLDDT scores. We then employ AlphaFold2 for protein structure prediction, followed by inputting these structures into AlphaFill (Hekkelman et al., 2023) to predict the associated ligands. If the predicted complex by AlphaFill includes the corresponding metal ion, we think the designed protein is highly potential to bind the metallocofactor. Our results indicate that all designed myoglobins are predicted to have a heme ligand, while 18 $\beta$-lactamases are predicted to have zinc ion ligand. **It shows that our method can generate proteins that are highly potential to bind the corresponding metallocofactors.**

**We also find our model is capable of designing diverse proteins.** We provide three designed $\beta$-lactamases in Figure 4 and find they have almost the same active site environments to those of B1, B2 and B3 metallo-$\beta$-lactamases reported in Palzkill (2013) (assessment process is provided in Appendix B.3). It validates that our method can design proteins with diverse structures.

**Our method is also able to design novel proteins.** We blast the 20 designed proteins in Uniprot and find two myoglobins that have low amino acid identity rates to the most similar one (PDB id=1SPG, Figure 5(c)) in database, say 66.0% and 26.7% respectively. It validates that our model has the ability to design novel protein sequences. Then we overlay their structures with 1SPG as shown in Figure 5 (a) and (b), of which (a) has almost the same structure (RMSD 0.458Å) and (b) has similar active site environments and is predicted to have heme ligand by AlphaFill but is slightly different with 1SPG (RMSD 3.943Å). It demonstrates the generated protein sequences by our model also have reasonable and novel structures. We provide the sequences for all above cases and more designed samples in Appendix D.

## 6 CONCLUSION

This paper proposes NAEPro, a method to co-design protein sequence and backbone structure based on automatically detected meaningful fragments. NAEPro consists of stacked novel NAELs, each of which includes a global attention sub-layer and a neighborhood equivariant sub-layer to facilitate comprehensive information exchange. Experimental results show that our NAEPro is able to design novel proteins with desired functions. One limitation of this work is, although our model demonstrats promising results, the designed proteins have not undergone wet-lab testing. Future work will involve the wet-lab testing to verify the biological functions of NAEPro designed proteins.

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

APPENDIX

# A    PROOF OF THEOREM 3.1 AND COROLLARY 3.2

## A.1    PROOF OF THEOREM 3.1

In this section we prove that our proposed NAEL is translation equivariant on $\boldsymbol{x}$ for any translation vector $\boldsymbol{t} \in \mathbb{R}^3$ and rotation equivariant for any $R$ from SO(3) group. More formally, we will prove the NAEL satisfies:

$$\boldsymbol{H}^{l+1}, R\boldsymbol{x}^{l+1} + \boldsymbol{t} = \text{NAEL}(\boldsymbol{H}^l, R\boldsymbol{x}^l + \boldsymbol{t}) \tag{12}$$

When $l = 0$:

For message update, the distance between two residues is invariant as $d_{ij}^l = ||R\boldsymbol{x}_i^l + \boldsymbol{t} - (R\boldsymbol{x}_j^l + \boldsymbol{t})||^2 = (\boldsymbol{x}_i^l - \boldsymbol{x}_j^l)^T R^T R (\boldsymbol{x}_i^l - \boldsymbol{x}_j^l) = ||\boldsymbol{x}_i^l - \boldsymbol{x}_j^l||^2$. $\boldsymbol{h}_i^l$ is the embedding of residue or [mask] token, and thus it is always invariant. Therefore, $\boldsymbol{h}_i^{l+0.5}$ is also invariant. Since $\boldsymbol{h}_i^{l+0.5}$, $\boldsymbol{h}_j^{l+0.5}$ and $d_{ij}^l$ are all invariant, and thus $\boldsymbol{m}_{ij}^{l+1}$ is also invariant to translation $\boldsymbol{t}$ and rotation $R$ on $\boldsymbol{x}$.

For coordinate update, updated $\boldsymbol{x}^{l+1}$ is equivariant to the translation $\boldsymbol{t}$ and rotation $R$ on input $\boldsymbol{x}^l$:

$$
\begin{aligned}
(R\boldsymbol{x}_i^l + \boldsymbol{t}) + &\sum_{j \in \text{Neighbor(i)}} (R\boldsymbol{x}_i^l + \boldsymbol{t} - (R\boldsymbol{x}_j^l + \boldsymbol{t})) \cdot \text{FFN}(\boldsymbol{m}_{ij}^{l+1}) \\
&= R(\boldsymbol{x}_i^l + \sum_{j \in \text{Neighbor(i)}} (\boldsymbol{x}_i^l - \boldsymbol{x}_j^l) \cdot \text{FFN}(\boldsymbol{m}_{ij}^{l+1})) + \boldsymbol{t} \\
&= R\boldsymbol{x}_i^{l+1} + \boldsymbol{t}
\end{aligned}
\tag{13}
$$

For residue update, $\boldsymbol{m}_{ij}^{l+1}$ and $\boldsymbol{h}_i^{l+0.5}$ is invariant to translation $\boldsymbol{t}$ and rotation $R$ on $\boldsymbol{x}$, so $\boldsymbol{h}_i^{l+1}$ is also invariant to translation $\boldsymbol{t}$ and rotation $R$ on $\boldsymbol{x}$.

When $l >= 1$:

We have proved that when $l = 0$, $\boldsymbol{h}_i^1$ is invariant to rotation and translation on $\boldsymbol{x}$. Taking $\boldsymbol{H}^1$ as the second layer input and following the above process, we can prove $\boldsymbol{h}_i^2$ is also invariant. Repeating this process from $l = 1$ to $L - 1$, we can get the same conclusion.

Combining the above two scenarios together, we have $\boldsymbol{H}^{l+1}, R\boldsymbol{x}^{l+1} + \boldsymbol{t} = \text{NAEL}(\boldsymbol{H}^l, R\boldsymbol{x}^l + \boldsymbol{t})$ for $l = 0$ to $L - 1$. Therefore, our proposed NAEL is $SE(3)$-equivariant.

## A.2    PROOF OF COROLLARY 3.2

We provide the proof of corollary 3.2 as follows:

*Proof.*

$$
\begin{aligned}
\text{NAEPro}(\boldsymbol{H}^0, R\boldsymbol{x}^0 + \boldsymbol{t}) &= \text{NAEL}^{L-1} \circ \text{NAEL}^{L-2} \circ \cdots \circ \text{NAEL}^0(\boldsymbol{H}^0, R\boldsymbol{x}^0 + \boldsymbol{t})) \\
&= \text{NAEL}^{L-1} \circ \text{NAEL}^{L-2} \circ \cdots \circ \text{NAEL}^1(\boldsymbol{H}^1, R\boldsymbol{x}^1 + \boldsymbol{t})) \\
&= ... \\
&= \text{NAEL}^{L-1}(\boldsymbol{H}^{L-1}, R\boldsymbol{x}^{L-1} + \boldsymbol{t})) = \boldsymbol{H}^L, R\boldsymbol{x}^L + \boldsymbol{t}
\end{aligned}
\tag{14}
$$

# B    ADDITIONAL EXPERIMENTAL DETAILS

## B.1    SIGNIFICANCE OF METALLOPROTEINS STUDIED HEREIN

Metalloproteins comprise almost 50% of all the naturally occuring proteins. The Zn-dependent $\beta$-lactamases and heme-dependent myoglobins studied herein represent biologically signicant metalloprotein examples. In particular, $\beta$-lactamases are enzymes produced by microorganisms to break

Figure 6: An example to illustrate that our model does not satisfy reflection equivariance. Here, left side sequence is Ala-Val which is made by natural amino acids. On the right side, the sequence is also Ala-Val. The reflection will not affect the sequence. However, the conformation of the amino acid did change from L-amino acid to D-amino acid.

| Protein | PDB | Metal Binding | Length Filtering |
|---|---|---|---|
| $\beta$-lactamase | 171, 484 | 7, 802 | 5, 427 |
| myoglobin | 14, 573 | 3, 381 | 3, 381 |

Table 3: Detailed data statistics of the two metalloproteins.

| Models | $\beta$-lactamase | myoglobin |
|---|---|---|
| Haluccination | 612.60 | 564.60 |
| Inpainting | 3.00 | 3.07 |
| SMCDiff+ProteinMPNN | 237.67 | 105.60 |
| PROTSEED | 4.87 | 3.05 |
| FrameDiff+ProteinMPNN | 124.95 | 68.65 |
| RFDiffusion+ProteinMPNN | 26.20 | 25.60 |
| NAEPro | **0.17** | **0.07** |

Table 4: Average inference time (s) for the two metalloproteins on test set.

down $\beta$-lactam antibiotics, conferring antibiotic resistance (Gupta, 2008). Thus, the study and design of $\beta$-lactamases hold relevance to public health and play a critical role in the development of new antibiotics. On the other hand, myoglobin is a heme-containing protein involved in oxygen storage and transport in muscle tissue (Springer et al., 1994), highlighting their biological significance. Note that the structure of hemoglobin closely resembles that of myoglobin (Antonini, 1965), so we merge these two metalloproteins and denote the merged dataset simply as myoglobin.

## B.2   PROTEIN DATA STATISTICS

Detailed data statistics for $\beta$-lactamase and myoglobin are reported in Table 3. We will release the two cleaned metalloprotein datasets in the near future.

## B.3   ACTIVE SITE ENVIRONMENT ASSESSMENT

We outline the procedures for verifying whether the designed metalloproteins share similar active site environments with their natural counterparts. For $\beta$-lactamase, we detect the residues that directly contact with zinc ion. If they show similar chemistry properties with natural $\beta$-lactamases, we think they can highly potentially bind to the corresponding metallocofactors. For myoglobin, we calculate the distance between axial histidine ligand to Fe ion. Besides, we also detect the presence of distal histidine within the active sites, which plays an important role in the natural function of myoglobin, i.e., oxygen molecule binding. If the distance is between 2.0Å and 2.5Å, and the distal histidine exists, we think the designed myoglobins are highly potentially to bind heme.

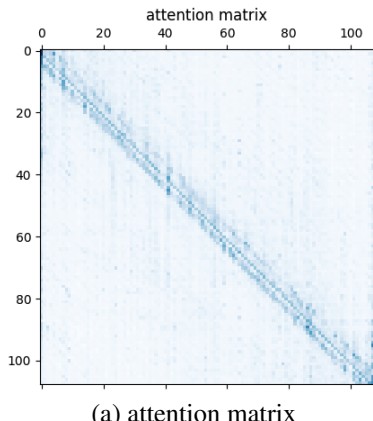
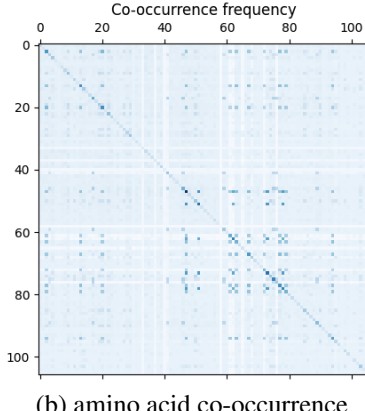

(a) attention matrix          (b) amino acid co-occurrence

Figure 7: Visualization of (a) attention matrix. (b) pairwise amino acid co-occurrence frequency.

| Case | Protein Sequence |
|---|---|
| Figure 4 (a) | EHSVVEISDDISITQLSDKVYTYVSLAEIEGWGMVPSNGMIVINNHQAALLDTPINDAQTEMLVNWVTDSLHAKVTTFIPNHWH GDCIGGLGYLQRKGVQSYANQMTIDLAKEKGLPVPEHGFTDSLTVSLDGMPLQCYYLGGGHATDNIVVWLPTENILFGGCMLK DNQTTSIGNISDADVTAWPKTLDKVKAKFPSARYVVPGHGNYGGTELIEHTKQIVNQYIESTS |
| Figure 4 (b) | GHSYEKYNNWETIEAWTKQVTSENPDLISRTAIGTTFLGNNIYLLKVGKPGPNKPAIFMDCGIHAREWISHAFCQWFVREAVLTY GYESHMTEFLNKLDFYVLPVLNIDGYIYTWTKNRMWRKTRSTNAGTTCIGTDPNRNFDAGWCTTGASTDPCDETYCGSAAESE KETKALADFIRNNLSSIKAYLSIHSYSQHIVYPYSYDYKLPENNAELNNLAKAAVKELATLYGTKYTYGPGATTLYLAPGGGDDW AYDQGIKYSFTFELRDKGRYGFILPESQIQATCEETMLAIKYVTNYVLGHLY |
| Figure 4 (c) | TVIKNETGTISISQLNKNVWVHTELGVPSNGLVLNTSKGLVLVDSSWDDKLTKELIEMVEKKFQKRVTDVIITHAHADHIGGIKTL KERGIKAHSTALTAELAKKNGYEEPLGDLQTVTNLKFGNMKVETFYPGKGHTEDNIVVWLPQYNILVGGSLVKSTSAKDLGNVV ADAYNEWSTSIENVLKRYRNINAVVPGHGEVGDKGLLLHTLDLLK |
| Figure 5 (a) | VDWTDAERAAITDLWAKVDVEDVGAQALARLLVVYPWTQRYFGGFGNISSASAILGNAKVAAHGKTVLTGLDRAIAHMDDIAG AFTQLSVKHSEKLHVDPDNFKVVGDLLTIVLAAVLGADFTPEVKAAWQKFLAVIVSALSRRYH |
| Figure 5 (b) | GFKQDIATIRGDLRTYAQDIFLAFLNKYPDERRYFKNYVGKSDQELKSMAKFGDHTEKVFNLMMEVADRATDCVPLASDANTLV QMKQHSSLTTGNFEKLFVALVEYMRASGQSFDSQSWDRFGKNLVSALSSAGMK |

Table 5: Protein sequences for cases in Figure 4 and 5.

## C ADDITIONAL EXPERIMENTS

### C.1 AVERAGE INFERENCE TIME

We provide detailed inference time of all models on the two datasets in Table 4. It shows our NAEPro has much faster inference speed than all other competitors. The reason is that our proposed NAEL update messages, residue coordinates and residue representations of all residues in an one-shot manner, leading to a much more efficient design process.

### C.2 ATTENTION VISUALIZATION

We randomly pick one sentence and provide the attention matrix visualization from the last layer in Figure **??**.

## D DESIGNED PROTEIN DISPLAY

### D.1 NOVEL AND DIVERSE PROTEIN SEQUENCES

We report the protein sequences shown in Figure 4 and 5 in Table 5.

### D.2 MORE DESIGNED CASES

Figure 8 and 9 respectively illustrate more designed $\beta$-lactamases and myoglobins. It shows all proteins exhibit active site environments reminiscent of those of natural ones and can bind to corresponding metallofactors, i.e., $\beta$-lactamases bind zinc ion and myoglobins bind heme, demonstrating their excellent potential to be biologically functional. Besides, these protein sequences are not exactly

the same as those in PDB, demonstrating our NAEPro has the ability to design novel proteins with desired functions.

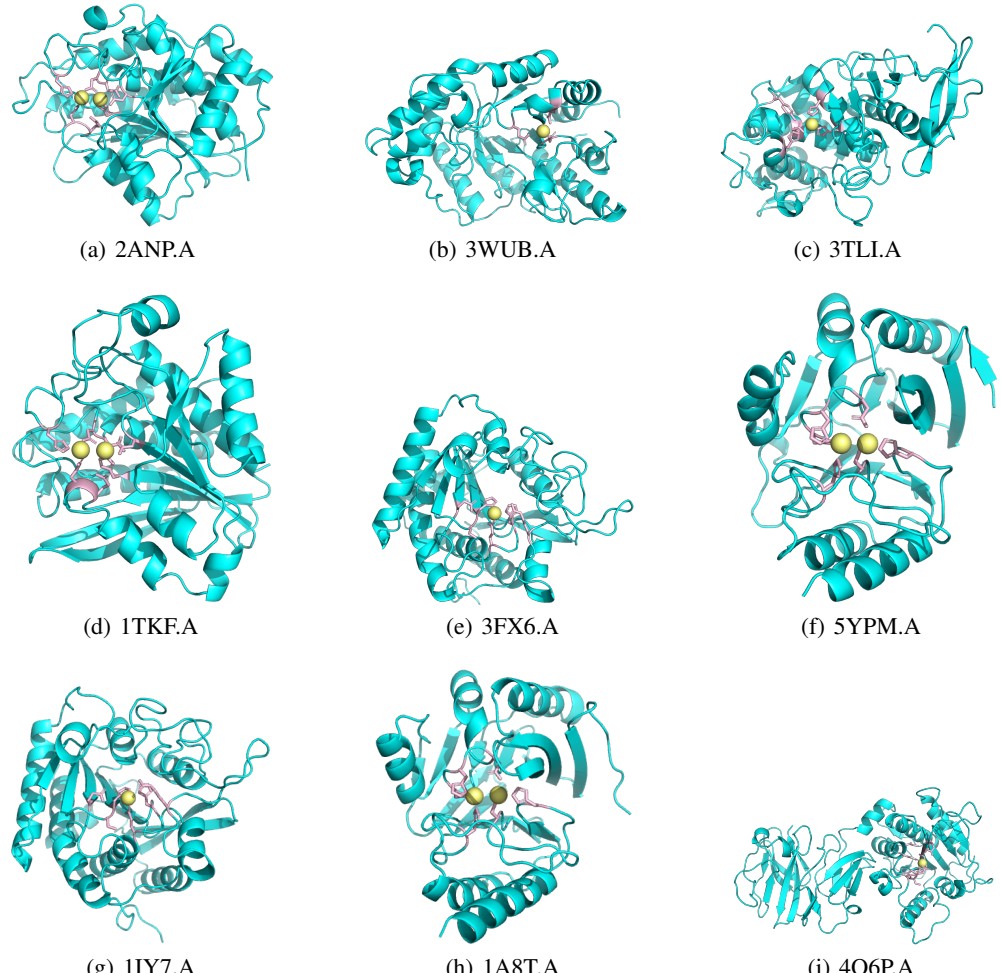

(a) 2ANP.A     (b) 3WUB.A     (c) 3TLI.A

(d) 1TKF.A     (e) 3FX6.A     (f) 5YPM.A

(g) 1IY7.A     (h) 1A8T.A     (i) 4Q6P.A

Figure 8: $\beta$-lactamases with motifs identical to those found in the target proteins in the PDB.

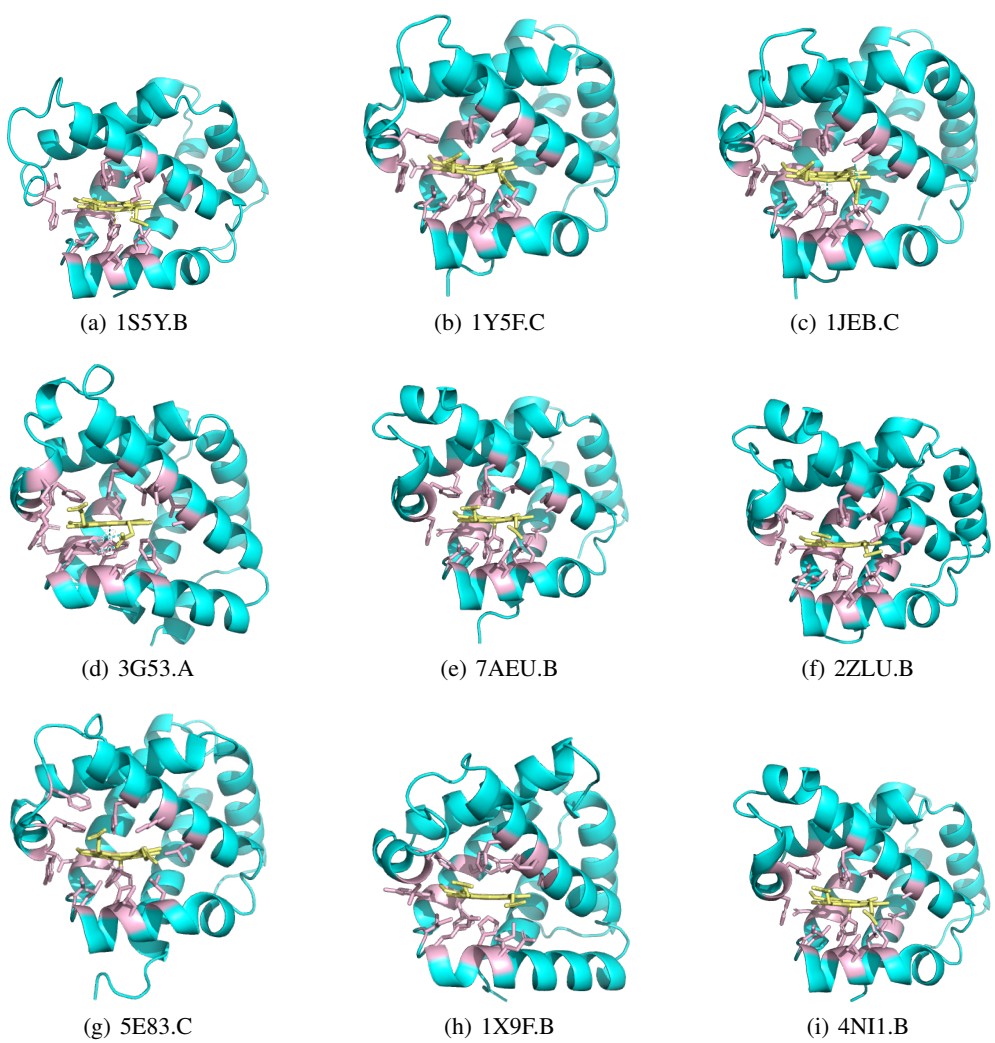

(a) 1S5Y.B

(b) 1Y5F.C

(c) 1JEB.C

(d) 3G53.A

(e) 7AEU.B

(f) 2ZLU.B

(g) 5E83.C

(h) 1X9F.B

(i) 4NI1.B

Figure 9: myoglobins with motifs identical to those found in the target proteins in the PDB.

