# OpenReview forum: "Functional Geometry Guided Protein Sequence and Backbone Structure Co-Design"
_ICLR.cc/2024/Conference — Submitted to ICLR 2024_

### Official Review · Reviewer_MgyU · 2023-10-20

**Soundness:** 3 good
**Presentation:** 3 good
**Contribution:** 2 fair
**Rating:** 6
**Confidence:** 5

**Summary:**

This manuscript proposed a transformer-like architecture for protein motif-scaffolding problem. Specifically, the protein motif, inferred from retrieved MSA profile, is input to the proposed model and both the sequence and structure of the rest region will be returned similar to the masked language models. The main contribution of this paper is the proposed network layer and the newly curated datasets, including beta-lactamase and myoglobin, for evaluating the scaffolding performance. Empirical results on the curated datasets show that the proposed method achieves better performance than baselines in several computational metrics.

**Strengths:**

- The author(s) proposed a new architecture for protein design w/ detailed introduction.
- The curated datasets have good domain significance for studying computational protein design/scaffolding problem.
- Ablation studies is performed for the proposed architecture.

**Weaknesses:**

Here are my concerns (w/ main questions):

- Regarding the consistency metrics reported, I noticed that the consistency metric of NAEPro in Table 1 (as the main results) underperformed half of the baselines. Since the consistency is defined by comparing the designed structure and predicted structure from designed sequence just similar to scTM/scRMSD, I wonder whether the proposed method captures sequence-structure cross-modality pattern through the mapping of NAEPro.  In particular, by jointly modeling both sequence and structure, one important strength to be expected from NAEPro is its capability of sequence-structure co-awareness and therefore generate more consistent proteins, compared with two-stage models such as RFdiffusion+IF in Table 1. However, such key observation showed conflict with this intuition and greatly weakened the method. The authors can elaborate on this point.
- For the experiments, how is the model trained to obtain the evaluation digits in Table 1 for each baseline. This is very important because as far as I acknowledged, all the baselines were simply pre-trained on general protein structure datasets such as PDB and CATH, plus the ProteinMPNN(IF). However, in the section 4.2, the author(s) mentioned that NAEPro, the proposed method, is trained on domain-specific dataset from beta-lactamase and myoglobin before inference stage. Do the author believe that such treatments for baseline comparison is fair? To contextualize, most of the protein design or engineering task (antibody/nanobody/enzyme; mutation effect) suffer from lack of domain-specific data, which therefore hinders the use of ML models. The presence of such specific dataset can have a significant effect on the performance of models. Therefore, the result shown in Table 1 may not as strong as it claimed.
- For the dataset, I found that in section 4.1, the author(s) mentioned that both beta-lactamase and myoglobin datasets are **split randomly.** This is worrying because the entries in PDB can usually be redundant, especially for such large and well-studied families that the authors selected for evaluation. Therefore, to prevent from data leakage, the sequence (even structure, since the structure-based metrics are also involved in this study) has to be redundant-removed by clustering or other strategy. Otherwise, highly similar (such as mutants) even the very same sequences and structures can present both in train and test dataset, making the result in Table 1 rather trivial. The most seemingly influenced metric is AAR, though it is itself questionable to reflect design models. However, due to the abnormally high AAR, other metrics can be biased more or less, due to the ESMFold-folded structures heavily rely on the single input sequence. The author(s) can elaborate on this point. Moreover, in the Table 3 (in appendix), I am curious about the column name “PDB”, does it mean the number of PDB entries (with unique ID) or the number of single chains? Any what is the sequence clustering statistics about the involved two metalloprotein datasets?
- For the evaluation metrics, the problem to be studied is functional protein (as it is claimed in the title) design instead of general backbone generation(eg., RFDiffusion) or sequence generation (eg., ProteinMPNN). Thus, I found the the five of the metrics fail to (or indirectly if they potentially do) reflect the performance of functional protein design or scaffolding. For the common practice of protein engineering, one may conduct wet-lab validation for the designed candidates (such as in ProteinMPNN) or use a fitness regression model to approximately indicate the success rate of the inference output. All these metrics in Table 1 & 2 show the somehow sequence-structure matching between the predicted and native, but are not very suitable for proteins such as enzyme.
- The potential theoretical flaw in the equivariance analysis in Section 3.5. To contextualize, the SE(3)-equivariance represents the equivariant property of functions with respect to elements in the special Euclidean group in R^3 which encompass only translation and rotation transformations (or roto-translation equivariance). Corresponding, the special orthogonal group mentioned in Theorem 3.1 also excludes the reflection matrices which has determinant equal to -1. However, in the Theorem 3.1., the author claimed SE(3)-equivariance yet let the proposed NAEL layer be rotation or reflection equivariant. Suppose it is a typo, however, assuming reflection-equivariant property can be problematic for protein structure, because the protein chirality can be important for helical structure.

**Questions:**

On top of the questions mentioned in the "Weakness" section, here are some further questions for the author(s) to answer:

- The proposed architecture encoding both sequence and structure via the masked language modeling (MLM) scheme is somehow interesting. How do the author(s) see the similarity difference between the proposed NAEL layer and tensor field convolutional layer in SE(3)-transformer[1]? For example, in the SE(3)-transformer implementation, one can also enable kNN message passing to save computational overhead. Also, what is the advantage of NAEL over the SeqIPA of the PROTSEED[2], the most competitive baseline in Table 1?
- In Section 3 - opening paragraph, the author(s) define/formulate the target task as generate a protein sequence and all 3D coordinates of N residues. However, in later model definition, the proposed NAEPro only operates on the C-alpha(CA) coordinates. Please explain this ambiguity. In fact, handling the all-atom coordinates and coarse-grained (CA-only) coordinates can have distinguishable scaling behavior/demand for the parameterized distribution / networks.
- In section 4.2, the author(s) described that the weights of NAEPro is partially initialized from pretrained ESM2-8M, of the smallest model size among the ESM2 series. This is very problematic but the recent common practices[3,4]  leverage at least the 650M model. Do the author(s) try much larger model size as initialization? I found in the Table 2 that the 8M model has only marginal improvement upon random initialization.
- How is the “target structure” in **evaluation metrics(Section 4.2)** determined? What is the difference between metrics RMSD and consistency?

[1] Fuchs, Fabian, Daniel Worrall, Volker Fischer, and Max Welling. "Se (3)-transformers: 3d roto-translation equivariant attention networks." *Advances in neural information processing systems* 33 (2020): 1970-1981.

[2] Shi, Chence, Chuanrui Wang, Jiarui Lu, Bozitao Zhong, and Jian Tang. "Protein sequence and structure co-design with equivariant translation." *arXiv preprint arXiv:2210.08761* (2022).

[3] Zheng, Zaixiang, Yifan Deng, Dongyu Xue, Yi Zhou, Fei Ye, and Quanquan Gu. "Structure-informed language models are protein designers." *bioRxiv* (2023): 2023-02.

[4] Verkuil, Robert, Ori Kabeli, Yilun Du, Basile IM Wicky, Lukas F. Milles, Justas Dauparas, David Baker, Sergey Ovchinnikov, Tom Sercu, and Alexander Rives. "Language models generalize beyond natural proteins." *bioRxiv* (2022): 2022-12.

---

> ### Author Response · Authors · 2023-11-17
> **Responses to reviewer MgyU: Part 1**
>
> We appreciate the reviewer’s valuable and insightful suggestions, which are very helpful for us to improve our paper. We have clarified all your concerns, added the experiments and updated the paper accordingly. We address the specific concerns as follows:
>
>  **Weakness 1: Regarding the consistency metrics reported, I noticed that the consistency metric of NAEPro in Table 1 (as the main results) underperformed half of the baselines.  Such key observation greatly weakened the method. The authors can elaborate on this point.**
>
> Ans: Yes, we agree with the reviewer’s opinion. We would like to clarify our task more clearly. Our goal is to design functional proteins with novel sequence and structure. As shown in Figure 5(b), the designed myoglobin has a RMSD of 3.943\AA with the most similar one in PDB and 26.7% AAR with the most similar one in Uniprot, but it can still bind the corresponding metallocofactor heme in the AlphaFill simulation [1]. In our paper, we use ESMFold to predict the structure of the designed sequence. However, as pointed out by reviewer VH6U, ESMFold might not accurately predict the structure for a novel sequence. To further clarify this point, we calculate the scRMSD of motifs between the designed structure and the predicted structure from the designed sequence in the following table. As the results show, our NAEPro achieves smaller RMSD than RFDiffusion+ProteinMPNN on myoglobin and comparable results on beta-lactamase.
>
> |Model| myoglobin|beta-lactamase|
> |:-----|:----:|:----:|
> |Inpainting|0.5292|0.7819|
> |SMCDiff+ProteinMPNN|1.3816|1.4523|
> |PROTSEED|0.4987|**0.3892**|
> |FrameDiff+ProteinMPNN|0.8765|0.9815|
> |RFDiffusion+ProteinMPNN|0.4936|0.3918|
> |NAEPro|**0.4832**|0.5239|
>
> [1] Alphafill: enriching alphafold models with ligands and cofactors. Maarten L Hekkelman et al. Nature methods. 2023.
>
> **Weakness 2: For the experiments, how is the model trained to obtain the evaluation digits in Table 1 for each baseline. This is very important.**
>
> Ans: We are sorry for the confusion. We got the codes of all baselines from their GitHub repos, and then re-trained all the models separately on myoglobin and beta-lactamase datasets. The evaluation procedure among all the baselines and our model follows the same process, and thus we believe the comparisons in Table 1 are fair.
>
> **Weakness 3: For the dataset, I found that in section 4.1, the author(s) mentioned that both beta-lactamase and myoglobin datasets are split randomly. This is worrying because the entries in PDB can usually be redundant, especially for such large and well-studied families that the authors selected for evaluation. The author(s) can elaborate on this point. Moreover, in the Table 3 (in appendix), I am curious about the column name “PDB”, does it mean the number of PDB entries (with unique ID) or the number of single chains?**
>
> Ans: Yes, this is a very good question. Since we focus on sequence and structure co-design task, we aim to design both functional sequence and structure. Therefore, our input data is the <sequence, structure> pair. From this point, even though the sequence might be the same, the structure is somehow different. Therefore, our paired input data are different and would not be redundant. We appreciate the reviewer for carefully reviewing our paper and also reading the appendix. In Table 3, PDB means the number of different PDB entries. To eliminate the reviewer’s concerns, we re-cleaned the test set by filtering pairs whose sequences have over 30% sequence identity to any sample in the training set after doing pairwise alignment for myoglobin. We compare our NAEPro and ProtSeed (the most competitive baseline) in the following table. As the results show, our model can design proteins with higher binding affinity (by Gnina [2]) on samples that have lower similarity to training data.
>
> |Model| RMSD|Binding Affinity (kcal/mol)|
> |:-----|:----:|:----:|
> |PROTSEED|**2.4031**|-9.62|
> |NAEPro|2.6634|**-9.98**|
>
> [2] GNINA 1.0: Molecular docking with deep learning. A McNutt et al. Cheminformatics. 2021.

---

> > ### Author Response · Authors · 2023-11-17
> > **Responses to reviewer MgyU: Part 2**
> >
> > **Weakness 4: For the evaluation metrics, the problem to be studied is functional protein (as it is claimed in the title) design instead of general backbone generation(eg., RFDiffusion) or sequence generation (eg., ProteinMPNN). Thus, I found the the five of the metrics fail to (or indirectly if they potentially do) reflect the performance of functional protein design or scaffolding. All these metrics in Table 1 & 2 show the somehow sequence-structure matching between the predicted and native, but are not very suitable for proteins such as enzyme.**
> >
> > Ans: In our analysis (Section 5.2), we evaluated the function for binding the corresponding metallocofactors of the designed proteins. Specifically, we first randomly select 20 cases from the top100 sequences (pLDDT ranking). We then employ AlphaFold2 for protein structure prediction, followed by inputting these structures into AlphaFill [1] to predict the associated ligands. Notably, our results indicate that all designed myoglobins exhibit heme binding capability, while 18 beta-lactamases demonstrate the ability to bind zinc ions, resulting in metallocofactor binding rates of 100% and 90%, respectively. It is evident that our method can generate proteins that express the basic and important functions, i.e. binding the corresponding metallocofactors. To systematically illustrate the function of our designed proteins, we use docking tool Gnina[2] to compute the binding affinity between the designed metalloproteins and the corresponding metallocofactors. The results in the following table shows that our model achieves the best binding affinity scores on both metalloproteins, demonstrating the proteins designed by our model are highly potential to actively exhibit biochemical functions.
> >
> > |Model| myoglobin (kcal/mol)|beta-lactamase (kcal/mol)|
> > |:-----|:----:|:----:|
> > |Hallucination|-7.23|-6.37|
> > |Inpainting|-9.32|-7.63|
> > |SMCDiff+ProteinMPNN|-8.79|-6.89|
> > |PROTSEED|-10.07|-7.68|
> > |FrameDiff+ProteinMPNN|-9.54|-7.21|
> > |RFDiffusion+ProteinMPNN|-9.76|-7.59|
> > |NAEPro|**-10.14**|**-7.80**|
> >
> > **Weakness 5: The potential theoretical flaw in the equivariance analysis in Section 3.5.**
> >
> > Ans: Thank the reviewer for pointing out this typo. Our model satisfies only roto-translation equivariance without reflection equivariance. We have accordingly updated the paper.
> >
> > **Q1: The proposed architecture encoding both sequence and structure via the masked language modeling (MLM) scheme is somehow interesting. How do the author(s) see the  difference between the proposed NAEL layer and tensor field convolutional layer in SE(3)-transformer ? Also, what is the advantage of NAEL over the SeqIPA of the PROTSEED, the most competitive baseline in Table 1?**
> >
> > Ans: The most biggest difference between our NAEPro and tensor field convolutional layer in SE(3)-transformer is:  NAEPro uses interleaving layers of global sequence-level attention and local neighborhood equivariant sub-layer, from which the residue representation is updated from both the whole sequence interaction and 3D neighbor message passing; instead, if we enable kNN message passing in SE(3)-transformer, then the node feature will only be updated based on 3D neighboring information. To clearly demonstrate the advantage of NAEL over SeqIPA in PROTSEED, we replace our NAEL with the SeqIPA-Addition without secondary structure feature. Taking myoglobin as an example, the results are shown in the following table. NAEPro with NAEL performs better than with SeqIPA.
> >
> > |Model| AAR (%)|RMSD|
> > |:-----|:----:|:----:|
> > |NAEPro -w/- SeqIPA|65.33|2.7078|
> > |NAEPro|**89.37**|**2.6307**|
> >
> > **Q2:  In Section 3 - opening paragraph, the author(s) define/formulate the target task as generate a protein sequence and all 3D coordinates of N residues. However, in later model definition, the proposed NAEPro only operates on the C-alpha(CA) coordinates. Please explain this ambiguity.**
> >
> > Ans: We are sorry for the ambiguity caused. When we mention “we formulate the target task as generate a protein sequence and all 3D coordinates of N residues”, we mean 3D coordinates of alpha-carbon of all the residues. Following the setting in [3], we represent the backbone structure as the CA-only coordinates of all residues. We have updated the paper accordingly in the new version.
> >
> > [3] Diffusion probabilistic modeling of protein backbones in 3D for the motif-scaffolding problem. Brian L. Trippe et al. ICLR. 2023.

---

> > > ### Author Response · Authors · 2023-11-17
> > > **Responses to reviewer MgyU: Part 3**
> > >
> > > **Q3: In section 4.2, the author(s) described that the weights of NAEPro is partially initialized from pretrained ESM2-8M, of the smallest model size among the ESM2 series. This is very problematic but the recent common practices leverage at least the 650M model. Do the author(s) try much larger model size as initialization? I found in the Table 2 that the 8M model has only marginal improvement upon random initialization.**
> > >
> > > Ans: Yes, we tried to partially initialize our model with 35M ESM2 (NAEPro-12) and also train this model on the merged myoglobin and beta-lactamase data (NAEPro-12-Unified). For myoglobin, enlarging the model to 12 layers will not influence the performance too much, but training the larger model on the merged dataset will increase the pLDDT and TM-score. For beta-lactamase, enlarging the model from 6 layers to 12 layers is beneficial and will increase the overall performance. Particularly, training the 12-layer model on merged dataset will significantly improve the performance of beta-lactamase, say improve the average TM-score from below 0.5 to above 0.5. Then we also partially initialize a 30-layer model with 150M ESM2 (NAEPro-30-Unified) and train the model on merged dataset. The performance for both myoglobin and beta-lactamase will be further improved.
> > >
> > > **myoglobin**
> > >
> > > |Model| AAR (%)|RMSD|pLDDT |TM-score |Parameters|
> > > |:-----|:----:|:----:|:----:|:----:|:----:|
> > > |NAEPro-6 |**89.37**|**2.6307**|81.9507|0.5692|11.6M|
> > > |NAEPro-12 |88.42|2.6333|82.2849|0.5673|52M|
> > > |NAEPro-12-Unified|86.81|2.6443|82.9736|0.5736|52M|
> > > |NAEPro-30-Unified|86.62|2.6410|**83.0173**|**0.5749**|223.7M|
> > >
> > > **$\beta$-lactamase**
> > >
> > > |Model| AAR (%)|RMSD|pLDDT |TM-score |Parameters|
> > > |:-----|:----:|:----:|:----:|:----:|:----:|
> > > |NAEPro-6 |65.71|2.9916|66.8705|0.4812|11.6M|
> > > |NAEPro-12 |67.28|2.9864|68.7394|0.4919|52M|
> > > |NAEPro-12-Unified|70.93|**2.9707**|69.9333|0.5055|52M|
> > > |NAEPro-30-Unified|**72.60**|2.9781|**71.0429**|**0.5089**|223.7M|
> > >
> > > **Q4: How is the “target structure” in evaluation metrics(Section 4.2) determined? What is the difference between metrics RMSD and consistency?**
> > >
> > > Ans: Target structure means the natural protein structure provided in PDB. RMSD is calculated as the RMSD between the designed protein structure and the one given in PDB. Consistency is calculated as the RMSD between the designed structure and the one predicted by ESMFold for the designed sequence.
> > >
> > > We really hope our responses address your concerns. If you have any other questions, we are very happy to continue discussions!

---

> > > > ### Author Response · Authors · 2023-11-21
> > > > **Looking forward to any other questions!**
> > > >
> > > > Thanks for your valuable review and comments again. We have addressed all your concerns and questions.
> > > >
> > > > If you have any other questions, please feel free to discuss with us.

---

> > > > ### Comment · Reviewer_MgyU · 2023-11-21
> > > > **Response to Part III**
> > > >
> > > > 1. It is surprising to see scaling the model does not yield consistent performance gain. Could the authors help explain this behavior? Also, I noticed that the authors merge the two myoglobin and $\beta$-lactamase data for training newly in the rebuttal response. Is there any insight for doing such multi-task learning?
> > > > 2. Thanks for clarification. Could the authors briefly explain how do you construct the one-to-one mapping between the design structure and the target structure in PDB? Is it determined by retrieving the one in PDB with the smallest RMSD? As far as I know, there is no common practice to determine this for the task of protein design, and I appreciate any explanation that clues me in.

---

> > > > > ### Author Response · Authors · 2023-11-22
> > > > > **Follow-up responses**
> > > > >
> > > > > We much appreciate the reviewer’s responses and suggestions, which help a lot to improve our paper.  We have accordingly revised our paper, including Table 1, Table 2, Section 4.2-4.4 and Section 5.2. Our responses to the reviewer’s follow-up questions are provided as follows:
> > > > >
> > > > > **About the evaluation metrics: I think the binding affinity could better support the effectiveness of NAEPro and be more suitable for the task (functional protein design) rather than the accuracy metrics (RMSD, AAR, etc.). Do the authors agree on this or not?** C**ould the authors explain how the Binding Affinity is calculated for both model based on Gnina? Specifically, I am worried that the metrics is averaged among the test set similar to other metrics.  If so, I suggest that the authors report median or top-k mean as evaluation.**
> > > > >
> > > > > Ans: We completely agree with reviewer MgyU that the binding affinity is a better metric for functional protein design rather than the accuracy metrics. We revised Table 1 and 2 in our paper according to your suggestion. The binding affinity is calculated using Gnina as the average score on the test set. Thank the reviewer for this valuable suggestion. We provide  top-5, top-10, top-30 mean and variance as well as the median in the following table and also revise our paper accordingly. From the results, we can see that our model achieves the best median and top-K binding affinity scores on both datasets.
> > > > >
> > > > > **Myoglobin**
> > > > >
> > > > > |Model| top-5 (kcal/mol)|top-10 (kcal/mol)|top-30 (kcal/mol)| median|
> > > > > |:-----|:----:|:----:|:----:|:----:|
> > > > > |Hallucination|-8.18$\pm$ 0.01|-8.07$\pm$0.03|-7.97$\pm$0.23|-7.25|
> > > > > |Inpainting|-13.47$\pm$0.02|-13.12$\pm$0.12|-12.31$\pm$0.54|-9.56|
> > > > > |SMCDiff+ProteinMPNN|-11.37$\pm$0.03|-11.12$\pm$0.31|-10.87$\pm$0.42|-8.76|
> > > > > |PROTSEED|-13.21$\pm$0.13|-12.89$\pm$0.42|-11.98$\pm$0.52|-10.23|
> > > > > |FrameDiff+ProteinMPNN|-13.13$\pm$0.05|-12.92$\pm$0.16|-12.21$\pm$0.23|-10.08|
> > > > > |RFDiffusion+ProteinMPNN|-13.68$\pm$0.02|-13.03$\pm$0.21|-12.56$\pm$0.43|-10.15|
> > > > > |NAEPro|**-14.12$\pm$0.01**|**-13.85$\pm$0.10**|**-13.06$\pm$0.38**|**-10.74**|
> > > > >
> > > > > **Beta-lactamase**
> > > > >
> > > > > |Model| top-5 (kcal/mol)|top-10 (kcal/mol)|top-30 (kcal/mol)| median|
> > > > > |:-----|:----:|:----:|:----:|:----:|
> > > > > |Hallucination|-6.98$\pm$0.01|-6.87$\pm$0.02|-6.69$\pm$0.05|-6.29|
> > > > > |Inpainting|-9.89$\pm$0.03|-9.54$\pm$0.16|-9.13$\pm$0.43|-7.24|
> > > > > |SMCDiff+ProteinMPNN|-9.10$\pm$0.01|-9.05 $\pm$0.02|-8.98$\pm$0.01|-6.97|
> > > > > |PROTSEED|-9.88$\pm$0.21|-9.51$\pm$0.41|-9.01$\pm$0.62|-7.31|
> > > > > |FrameDiff+ProteinMPNN|-9.54$\pm$0.03|-9.56$\pm$0.23|-8.89$\pm$0.35|-7.03|
> > > > > |RFDiffusion+ProteinMPNN|-9.87$\pm$0.05|-9.56$\pm$0.23|-9.12$\pm$0.53|-7.51|
> > > > > |NAEPro|**-10.06$\pm$0.05**|**-9.79$\pm$0.10**|**-9.39$\pm$0.12**|**-7.66**|
> > > > >
> > > > > **Corrected statement: ESMFold might not accurately predict the structure for a novel sequence.**
> > > > >
> > > > > Ans: Sorry for the misunderstood claim. We want to express we agree on reviewer VH6U’s comment in question 6 “for a novel sequence, its prediction is worse than AlphaFold2.”  As suggested by reviewer VH6U, we re-calculate pLDDT by AlphaFold2. Our model respectively achieves 91.4139 and 76.9550  on myoglobin and $\beta$-lactamase on average. As claimed in previous work [1,2], pLDDT between 70~90 are classified to be confident and pLDDT ≥ 90 indicates residues predicted with extremely high confidence.
> > > > >
> > > > > [1] pLDDT Values in AlphaFold2 Protein Models Are Unrelated to Globular Protein Local Flexibility. Oliviero Carugo. Crystals. 2023.
> > > > >
> > > > > [2] Generating new protein sequences by using dense network and attention mechanism. Feng Wang et al. Mathematical Biosciences and Engineering. 2023.

---

> > > ### Comment · Reviewer_MgyU · 2023-11-21
> > > **Response to Part II**
> > >
> > > 1. I appreciate the complemented benchmarking experiments. On top of my previous response (Part I), I think the binding affinity or something could better support the effectiveness of NAEPro and be more suitable for the task (functional protein design) rather than the accuracy metrics (RMSD, AAR, etc.). Do the authors agree on this or not? Even though docking is absolutely not an ideal validation method in my opinion, but it is relatively convincing.
> > >     - For the claim “binding rates of 100% and 90%”, I wonder how is the criterion/threshold of “binding” determined? This success rate is rather unrealistic and probably fails to reflect the real performance. Thus, the statement in Section 5.2. saying “It is evident that our method can
> > >     generate proteins that express the basic and important functions.” can be quite misleading for readers who are not familiar with the task.
> > > 2. Great. Could the author intuitively describe how does the NAEL deal with the reflection case so as to be simply SE(3)-equivariant instead of E(3)? Since I found the update in NAEL share many merits with the Equivariant Graph Convolutional Layer (EGCL) in EGNN [1], where any orthogonal matrix can be applied and thus the E(3)-equivariance is achieved.
> > > 3. I appreciate the ablation. The authors are encouraged to encompass appropriate comparison with EGNN/SE(3)-transformer in their writing to better contextualize the proposed NAEL.
> > > 4. Good. I raised this concern because the “all-atom” for protein structure does refer to all the atoms (at most excluding hydrogen) literally instead of the coarse-grained structure (modeling only CA or CB). This can be misleading.
> > >
> > > [1] Satorras, V.G., Hoogeboom, E. and Welling, M., 2021, July. E (n) equivariant graph neural networks. In *International conference on machine learning* (pp. 9323-9332). PMLR.

---

> > ### Comment · Reviewer_MgyU · 2023-11-21
> > **Response to Part I**
> >
> > First of all, I appreciate the effort and active response of the authors. Here are some further comments, accordingly:
> >
> > 1. I understand that the ultimate goal is to “**design functional protein”** (sequence, structure) for some specific protein class such as $\beta$-lactamase. But however, the authors adopt the structure/sequence-based accuracy metrics (consistency, scRMSD, etc.) to benchmark the proposed method. Could you please explain the purpose and status of these metrics since they cannot reflect the actual functionality? Also, I have several points related to this question as follows:
> >     - The authors mentioned that "ESMFold might not accurately predict the structure for a novel sequence.", which was raised by reviewer VH6U. However, I am afraid this is a garbling statement, I only found that VH6U delineated AF2/ESMFold as "…cannot predict structures for all proteins with no error, it's a **great reference…**".
> >     - Moreover, “but it can still bind the corresponding metallocofactor heme in the AlphaFill simulation”. This statement is not convincing to me: in silico validations, even the computationally intensive flexible docking, can only somehow probe the behavior for the designed species. The statement above implied that RMSD as high as 3.943A while 26.7% AAR can still be good binder, which weakens any conclusion deduced from the benchmark results using RMSD/AAR, w.r.t. the functional protein design task.
> >     - Lastly, since the authors finally use the docking pipeline like Gnina and AlphaFill as the validation of binding and showing success rate for both tasks, why not using such metric to benchmark the baseline models? Seemingly it could be more suitable for the task “**design functional protein”.**
> > 2. Great! That makes sense to me.
> > 3. In the context of structure prediction (as well as what I put in the review above), what “redundant” mean can be beyond “identical” but something related to “homology”. Since the benchmark is heavily based on the accuracy metrics, I believe the redundancy-removal (such as 30%) is necessary. For the new result, could the authors explain how the **Binding Affinity(kcal/mol)** is calculated for both model based on Gnina? Specifically, I am worried that the metrics is averaged among the test set similar to other metrics. However, in design practice, the “mean” can be useless since only the top candidates are selected for the next stage. If so, I suggest that the authors report median or top-k mean as evaluation.

---

> ### Author Response · Authors · 2023-11-22
> **Follow-up responses: part2**
>
> **Corrected statement: But it can still bind the corresponding metallocofactor heme in the AlphaFill simulation. For the claim “binding rates of 100% and 90%”, I wonder how is the criterion/threshold of “binding” determined? This success rate is rather unrealistic and probably fails to reflect the real performance. Thus, the statement in Section 5.2. saying “It is evident that our method can generate proteins that express the basic and important functions.” can be quite misleading for readers who are not familiar with the task.**
>
> Ans: Sorry for the misleading statement. We mean through the AlphaFill prediction, the designed proteins are highly potential to bind the corresponding metallocofactors. We have corrected all the claims in our paper. We provided the process of how to decide if the designed protein can “bind” the metallocofactors in Appendix B.3. We would like to further explain it here.  First of all, we use AlphaFill to do the first-step prediction. AlphaFill will return a complex based on the input protein structure. If the complex includes the corresponding metal ions, we think the designed protein is potential to have functions. Then to guarantee the results convincing, we further use some additional constraints to ensure the designed proteins are potential to express functions. Specifically, we assume if a designed protein share the similar active site environments with their natural counterparts (target protein in our dataset), then it’s highly potential to bind the corresponding metal ions. For myoglobin, we calculate the distance between axial histidine ligand to Fe ion. Besides, we also detect the presence of distal histidine within the active sites, which plays an important role in the oxygen molecule binding function of myoglobin . If the distance is between 2.0Å and 2.5Å, and the distal histidine exists, we think the designed myoglobins share similar active site environments to natural ones. For $beta$-lactamase, we detect the residues that directly contact with zinc ion. If they show similar chemistry properties with natural $beta$-lactamases, we think they can highly potentially bind the corresponding metallocofactors. For example, in Figure 5(a), the active site possesses one zinc atom coordinated to three histidines, while the second zinc atom is coordinated to one histidine, one cysteine, and one aspartate, demonstrating a high degree of amino acid analogy to the natural protein.
>
> **Could the author intuitively describe how does the NAEL deal with the reflection case so as to be simply SE(3)-equivariant instead of E(3)?**
>
> Ans: Intuitively, applying reflection to the input structure, the output structure will also be reflected. The conformation of the original backbone will be changed from L-amino acid to D-amino acid (We provide an illustration in Figure 6 in appendix). Therefore, the reflection will change the chirality of the protein which may cause the deficiency of binding to the ligand and eventually its function. However, the protein sequence in our model will not be influenced. Therefore, the 3D structure of the designed sequence which is L-amino acid and the output structure  which becomes D-amino acid will be inconsistent. From this aspect, our model doesn’t satisfy reflection equivariance.
>
> **The authors are encouraged to encompass appropriate comparison with EGNN/SE(3)-transformer in their writing to better contextualize the proposed NAEL.**
>
> Ans: We provide the comparison with EGNN+ESM2 on myoglobin in the following Table and also updated the results in Table2 accordingly.
>
> |Model| top-5 (kcal/mol)|top-10 (kcal/mol)|top-30 (kcal/mol)| median|
> |:-----|:----:|:----:|:----:|:----:|
> |EGNN+ESM2|-13.30$\pm$0.61 | -12.77$\pm$0.69 |-12.07$\pm$0.65 | -9.63|
> |NAEPro|**-14.12$\pm$0.01**|**-13.85$\pm$0.10**|**-13.06$\pm$0.38**|**-10.74**|
>
> **It is surprising to see scaling the model does not yield consistent performance gain. Could the authors help explain this behavior? Also, I noticed that the authors merge the two myoglobin and lactamase data for training newly in the rebuttal response. Is there any insight for doing such multi-task learning?**
>
> Ans: Enlarging the model hasn’t improved the performance too much. Our interpretation is that the training data size is moderate, and thus a model with moderate size can fit the data well. To validate if this guess holds, we merge the two protein data and then train a unified model. As we can see in the previous table, our model did achieve obvious better results on $\beta$-lactamase, i.e., TM-score from below 0.5 (0.4919) to over 0.5 (0.5055).

---

> > ### Author Response · Authors · 2023-11-22
> > **Follow-up responses: part3**
> >
> > **Could the authors briefly explain how do you construct the one-to-one mapping between the design structure and the target structure in PDB? Is it determined by retrieving the one in PDB with the smallest RMSD? As far as I know, there is no common practice to determine this for the task of protein design, and I appreciate any explanation that clues me in.**
> >
> > Ans: Yes, we construct the one-to-one mapping between the designed structure and the target structure in PDB. It is not determined by retrieving the one in PDB with the smallest RMSD. We extract the motif fragments for each protein, based on which our training objective is to recover the original protein sequence and backbone structure. Therefore, we compute the RMSD between the designed structure and the target one which we want to recover.

---

> > > ### Comment · Reviewer_MgyU · 2023-11-22
> > > **Response to the authors and summary**
> > >
> > > Dear Authors,
> > >
> > > Thanks for your responsive and polite discussion as well as the dedicated rebuttal. I am satisfied with what you have addressed now. I have accordingly adjusted the score and evaluations as we had good and nice communication. I hope our discussion did improve the quality/clarity of the manuscript and can be beneficial.
> > >
> > > To finalize the review for AC, this paper is recognized with clear presentation, meticulous ablation study, and state-of-the-art performance on the revised benchmark. However, a notable limitation lies in the absence of deeper insights (or “novelty” as other reviewers put), as in silico validation alone is not sufficient to fully support the relevant conclusion. While the attempt is commendable, a more comprehensive study for the "protein design" task would add weight to the paper's overall contribution to the relevant field.
> > >
> > > Regards

---

> > > > ### Author Response · Authors · 2023-11-22
> > > > **Response to reviewer MgyU**
> > > >
> > > > Dear reviewer,
> > > >
> > > >     We sincerely appreciate the time and effort you dedicated to reviewing our paper and reading our responses. Your valuable suggestions and insightful comments have significantly contributed to refining the quality of our work. No matter what the final result will be, the thoughtful communication with you has better clarified our paper's goals and logic. We would like to express our great gratitude to you!
> > > >
> > > > Best,
> > > >
> > > > Authors of paper 4608

---

### Official Review · Reviewer_Wq6M · 2023-11-01

**Soundness:** 2 fair
**Presentation:** 3 good
**Contribution:** 2 fair
**Rating:** 3
**Confidence:** 5

**Summary:**

This paper proposes NAEPro, a model to jointly design Protein sequence and structure. NAEPro is powered by an interleaving network of attention and equivariant layers, which can capture global correlation in a whole sequence and local influence from the nearest amino acids in three-dimensional (3D) space. The global attention sub-layer parameters are initialized with ESM-2. The author combines ESM2 and EGNN for co-modeling protein sequence and structure.

**Strengths:**

1. The reported performance is good.
2. The method is simple.

**Weaknesses:**

1. Novelty: Both EGNN and ESM2 are existing models, and there are many existing works on antibody structure and sequence co-design. The combination may limit the novelty of this method.
2. Significance: It is likely that previous works can be readily applied to the motif-conditioned setting, and the authors can easily adapt their method to antibody design tasks. It would be beneficial if the authors further elaborate on the significance of their work
3. Code: The authors do not provide code for checking the soundness of the methods.
4. Experiment setting: The authors do not provide results on standard benchmarks, such as the CATH dataset, for fair comparison on both sequence and structure design.

**Questions:**

1. Could you provide experimental results from the CATH dataset to compare with the original SMCDiff and FrameDiff results?
2. Similarly, could you provide head-to-head comparisons to ProteinMPNN, ESMIF, and PiFold on protein sequence design? Please follow the same setting.
3. Could you provide the code for checking the results?

---

> ### Author Response · Authors · 2023-11-17
> **Responses to reviewer Wq6M: Part 1**
>
> We thank the reviewer for the positive reviews as well as the suggestions for improvement. We have clarified all your concerns, added the experiments and updated the paper accordingly. Our responses to the reviewer’s concerns and questions are provided below:
>
> **Weakness 1: Both EGNN and ESM2 are existing models, and there are many existing works on antibody structure and sequence co-design. The combination may limit the novelty of this method.**
>
> Ans: We would like to point out that our NAEPro is not a combination of ESM2 and EGNN. Instead our key innovations are (1) interleaving layers of sequence-level attention and local neighborhood equivariant sub-layer (2) accelerating the local sub-layer with the k-nearest neighbors.  On the contrary,  EGNN is a fully-connected graph and updates message, coordinates and atom features based on all other atoms in 3D space, which is not efficient for long proteins. Besides, EGNN+ESM2 is a sequential process and can not cross-condition on sequence and structure. We compare our method with EGNN+ESM2, NAEPro w/o ESM2 initialization on myoglobin in the Following Table (Table 2 in the paper).  As the results show, EGNN+ESM2 performs much worse than our NAEPro, and removing the ESM2 initialization will not influence the performance too much, demonstrating that our model is much more superior than EGNN+ESM2.
>
> |Model| AAR (%)|RMSD|pLDDT |TM-score |consistency|
> |:-----|:----:|:----:|:----:|:----:|:----:|
> |EGNN+ESM2|51.12|2.9891|77.3399|0.4656|4.9827|
> |NAEPro-w/o-ESM2 initialization|79.82|2.6398|76.3032|0.5159|4.7273|
> |NAEPro|**89.37**|**2.6307**|**81.9507**|**0.5692**|**4.3865**|
>
> **Weakness 2: It is likely that previous works can be readily applied to the motif-conditioned setting, and the authors can easily adapt their method to antibody design tasks. It would be beneficial if the authors further elaborate on the significance of their work**
>
> Ans: We would like to clarify our goal more clearly. Our task is to co-design functional and novel protein sequence and structure. Previous work which is most similar to ours is Inpainting [1]. However, [1] provides the functional sites manually while we automatically mined the functional sites (plus the conserved sites as suggested by reviewer ****VH6U****) by MSAs. Previous methods considering only sequence design constrained by fitness value [2,3] or sequence design based on given backbone structure (inverse folding) [4] is not suitable for our task. For antibody design, antibody is always Y-shaped, while one of our goal is to design novel and diverse protein structure. For example, in Figure 4, the designed beta-lactamases have different fold categories and belong to three different sub-classes. In Figure 5 (b), the designed myoglobin has a RMSD of 3.943\AA with the most similar one in PDB, but it can still bind the corresponding metallocofactor heme in the AlphaFill simulation [5]. From this aspect, we think antibody design models can not be directly applied to our task without any modification and vice versa.
>
> [1] Scaffolding protein functional sites using deep learning.  Jue wang. et al. Science. 2022.
>
> [2] Biological Sequence Design with GFlowNets. [Moksh Jain](https://arxiv.org/search/q-bio?searchtype=author&query=Jain,+M) et al. ICML 2022.
>
> [3] Proximal Exploration for Model-guided Protein Sequence Design. Zhizhou Ren et al. ICML 2022.
>
> [4] Robust deep learning based protein sequence design using ProteinMPNN. J. Dauparas et al. Science. 2022.
>
> [5] Alphafill: enriching alphafold models with ligands and cofactors. Maarten L Hekkelman et al. Nature methods. 2023.
>
> **Weakness 3:** **The authors do not provide code for checking the soundness of the methods.**
>
> Ans: We have provided the code and data in the supplementary material. Due to the memory limitation, we couldn’t upload the model checkpoints. However, the training process is highly efficient, and the reviewer could train the model if he/she is interested.

---

> > ### Author Response · Authors · 2023-11-17
> > **Responses to reviewer Wq6M: Part 2**
> >
> > **Weakness4 & Q1: The authors do not provide results on standard benchmarks, such as the CATH dataset, for fair comparison on both sequence and structure design. Could you provide experimental results from the CATH dataset to compare with the original SMCDiff and FrameDiff results?**
> >
> > Ans: We would like to clarify our task more clearly. Our task aims to co-design functional and novel protein sequence and structure. In our method, we use MSAs to automatically find motifs to guarantee the protein function, which is achieved within the same protein family. Instead, CATH is a protein dataset consisting of proteins from diverse families, making the motif extraction difficult. Besides, CATH is usually used to evaluate inverse folding task, say protein sequence design based on fixed backbone structure, which is not the goal of our paper. We hope we didn’t misunderstand the reviewer’s meaning by “the original SMCDiff and FrameDiff results” as SMCDiff and FrameDiff haven’t been evaluated on CATH dataset. Since SMCDiff and FrameDiff focus on protein structure design, CATH is usually used to evaluate protein sequence design based on fixed backbone structure, which may have some discrepancies on the design goals.
> >
> > **Q2: Could you provide head-to-head comparisons to ProteinMPNN, ESMIF, and PiFold on protein sequence design? Please follow the same setting.**
> >
> > Ans: We provide protein sequence design given backbone structures on our two datasets in the following table. In this setting, we slightly modified our method to make it adaptable to the task as the reviewer asked. Specifically, we provided the whole backbone structure as the model input and designed the whole protein sequence. As the results show, even though we only provide  backbone structure without any motif residues, our model can still achieves higher AAR and pLDDT on myoglobin, and higher AAR on beta-lactamase.
> >
> > **myoglobin**
> >
> > |Model| AAR (%)|pLDDT |
> > |:-----|:----:|:----:|
> > |ProteinMPNN|81.37|81.9817|
> > |ESMIF|76.49|78.0986|
> > |PiFold|78.93|80.2912|
> > |NAEPro|**85.56**|**82.3871**|
> >
> > $\beta$-lactamase
> >
> > |Model| AAR (%)|pLDDT |
> > |:-----|:----:|:----:|
> > |ProteinMPNN|53.59|**70.6723**|
> > |ESMIF|57.49|61.3928|
> > |PiFold|63.38|64.9017|
> > |NAEPro|**68.39**|65.8218|
> >
> > **Q3: Could you provide the code for checking the results?**
> >
> > Ans: We have provided the code and data in the supplementary material. Due to the memory limitation, we couldn’t upload the model checkpoints. However, the training process is highly efficient, and the reviewer could train the model if he/she is interested.

---

> > > ### Author Response · Authors · 2023-11-21
> > > **Looking forward to any other questions**
> > >
> > > Thanks for your valuable review and comments again. We have addressed all your concerns and questions.
> > >
> > > If you have any other questions, please feel free to discuss with us.

---

> > > ### Comment · Reviewer_Wq6M · 2023-11-22
> > > **Response to authors**
> > >
> > > **About Weakness4 & Q1** Page 2, "We carry out experiments on two metalloproteins, including β-lactamase and myoglobin." I would like to say that the experimental setting is the main concern to me. Since the authors have only evaluated the proposed method on two two structures, I am not sure if the method can be extended to generalized situations. I know that you are doing protein sequence-structure inpaintning problem, in which case a comparison with a previous baselines [1,2,3] on stantard datasets may help to reveal the generalization capabilities of your algorithm.
> > >
> > > [1] Trippe, Brian L., et al. "Diffusion probabilistic modeling of protein backbones in 3d for the motif-scaffolding problem." arXiv preprint arXiv:2206.04119 (2022).
> > >
> > > [2] Lee, Jin Sub, Jisun Kim, and Philip M. Kim. "Score-based generative modeling for de novo protein design." Nature Computational Science (2023): 1-11.
> > >
> > > [3] Watson, Joseph L., et al. "De novo design of protein structure and function with RFdiffusion." Nature 620.7976 (2023): 1089-1100.
> > >
> > > **About Q2** Your methods are not carefully designed for protein inverse folding. However, the presented results outperformed current SOTA by a large margin (curent SOTA is about 50%-60% AAR), which could not convince me. I doubt the veracity and correctness of the experimental results.

---

> ### Comment · Reviewer_Wq6M · 2023-11-22
> **Response to authors**
>
> Thanks for the authors' efforts. I have additional concerns:
>
> - Page 1, "Despite their great potential for novel structure design, such sequential design policy fails to cross-condition on sequence and structure, which might lead to inconsistent proteins and inefficient design process". I can not agree with this statement. RFDiffusion's authors say that they also considered simultaneously designing structure and sequence within RFdiffusion, but combining ProteinMPNN with the diffusion of structure alone provides the excellent performance, as shown in your Table.1. **Unfortunately, I observe that you deleted the consistency metric in Table.1,which seems like dishonest behavior to me.**
>
> - Page 4, "Through this way, amino-acid combinations frequently occur in the same context would draw higher attention scores". Have you checked this statement? Could you provide some visualizations of the attention map and correlated frequences of amino-acid combinations?
>
> - Page 4, "Updating residue representations and coordinates in 3D space with only nearest neighbors enables more efficient and economic message passing compared to prior approaches which compute messages on the complete pairwise residue graph". To my knowledge, most of the previous methods such as proteinmpnn adopt knn for constructing sparse graph in the 3D spcace. You should provide evidence to support the significance of your statement.
>
> - Novelty: I do not observe enough novelty from the perspective of machine learning in algorithm design.

---

> ### Author Response · Authors · 2023-11-23
> **Follow-up responses to reviewer Wq6M**
>
> Thanks for the responses! Our answers to your follow-up questions are provided as follows:
>
> **1. About Weakness4 & Q1 Page 2, "We carry out experiments on two metalloproteins, including β-lactamase and myoglobin." I would like to say that the experimental setting is the main concern to me. Since the authors have only evaluated the proposed method on two structures, I am not sure if the method can be extended to generalized situations. I know that you are doing protein sequence-structure inpaintning problem, in which case a comparison with a previous baselines [1,2,3] on stantard datasets may help to reveal the generalization capabilities of your algorithm.**
>
> Ans: We agree with reviewer Wq6M and additionally compare our method with PROTSEED on B12 protein. Due to the approaching ddl and limited time, PROTSEED has only been trained for 100 epochs, the same as our method. (suggested training epoch is 2000 on their GitHub)
>
> The results are reported as follows:
>
> **Binding affinity score for designed B12**
>
> |Model| top-5 (kcal/mol)|top-10 (kcal/mol)|top-30 (kcal/mol)| median|
> |:-----|:----:|:----:|:----:|:----:|
> |PROTSEED|-9.45$\pm$ 0.03|-9.36$\pm$0.09|-9.04$\pm$0.27|-8.46|
> |NAEPro|**-11.06$\pm$0.14**|**-10.96$\pm$0.14**|**-10.61$\pm$0.29**|**-9.11**|
>
> **2. About Q2 Your methods are not carefully designed for protein inverse folding. However, the presented results outperformed current SOTA by a large margin (curent SOTA is about 50%-60% AAR), which could not convince me. I doubt the veracity and correctness of the experimental results.**
>
> Ans: Sorry for the confusion. We evaluate the inverse-folding task on our own two datasets instead of the original CATH. For our datasets, the proteins are from the same family, and thus they may have much overlap on the sequence level, which definitely will lead to high AAR. Besides, our model is initialized with ESM2 weights, which will also improve the AAR.
>
> Although the sequence prediction based a fixed backbone structure is not our objective, **we adapt our method to CATH setting by keeping all the CA coordinates and masking all residues. Due to the approaching ddl and limited time, our model has only been trained for 10 epochs, while usually the suggested training epoch would be 100 like GVP, PiFold, etc.** (See below: ESMIF, ProteinMPNN, PiFold results are quoted from PiFold paper).
>
> CATH 4.2 results
>
> |Model| AAR (%)|PPL |
> |:-----|:----:|:----:|
> |ProteinMPNN|45.96|4.61|
> |ESMIF|38.30|6.44|
> |PiFold|51.66|4.55|
> |NAEPro|9.16|15.26|
>
> **Again, we did not claim the superiority of our method on inverse folding task on CATH dataset.**
>
> **3. Page 1, "Despite their great potential for novel structure design, such sequential design policy fails to cross-condition on sequence and structure, which might lead to inconsistent proteins and inefficient design process". I can not agree with this statement. RFDiffusion's authors say that they also considered simultaneously designing structure and sequence within RFdiffusion, but combining ProteinMPNN with the diffusion of structure alone provides the excellent performance, as shown in your Table.1. Unfortunately, I observe that you deleted the consistency metric in Table.1, which seems like dishonest behavior to me.**
>
> Ans: We do not intend to hide any information. As suggested by reviewer MgyU, the binding affinity is a better metric for functional protein design rather than the accuracy metrics (consistency, AAR, RMSD, etc.). Therefore, we replaced all the accuracy metrics in Table 1 and 2 in our paper according with the binding affinity scores calculated by Gnina. The full table with all scores are listed below:
>
> $\beta$-lactamase
>
> |Model| AAR (\%,$\uparrow$) | RMSD ({ \AA},$\downarrow$) | pLDDT ($\uparrow$) | TM-score ($\uparrow$) | Consistency (\AA,$\downarrow$)|
> |:-----|:----:|:----:|:----:|:----:|:----:|
> |Hallucination | $4.79$ | $--$ | $30.5511$ | $0.2918$ | $--$|
> |Inpainting | $16.73$ | $4.0599$ | $61.7679$ | $0.3790$ | $6.2578$|
> |SMCDiff+ProteinMPNN| $19.94$ | $10.3960$ | $42.0375$ | $0.3458$ |$10.2117$|
> | PROTSEED | $37.63$ | $3.0142$ | $64.3861$| $0.4637$ | $3.3748$ |
> | FrameDiff+IF |$26.20$ | $6.0151$ | $65.6445$ | $0.3657$ | $7.8703$|
> |RFDiffusion+IF | $22.93$ | $6.0438$ | **83.4058** | $0.3747$ |**0.5565**|
> |NAEPro|**65.71**| **2.9916**| 66.8705|**0.4812**| 7.3760|
>
>
> **Myoglobin**
> |Model| AAR (\%,$\uparrow$) | RMSD (\AA,$\downarrow$) | pLDDT ($\uparrow$) | TM-score ($\uparrow$) | Consistency (\AA,$\downarrow$)|
> |:-----|:----:|:----:|:----:|:----:|:----:|
> |Hallucination| $4.81$ | $--$ | $38.2817$ | $0.2754$ |--|
> |Inpainting |39.59|3.3751|67.0813|0.4391 |3.2108|
> |SMCDiff+IF | $12.47$ | $8.0067$ | $34.5914$ | $0.2235$ | $8.8754$ |
> | PROTSEED | $48.55$ | $2.8753$ | $61.5588$ | $0.5466$ | $0.9764$ |
> | FrameDiff+IF | $20.43$ | $5.9739$ | $61.3945$ | $0.3757$ |$3.5078$|
> |RFDiffusion+IF | $32.70$ | $3.9930$ | $78.9868$ | $0.4147$  | **0.4375** |
> |NAEPro | **89.37**| **2.6307**|**81.9507**|**0.5692**|4.3865|

---

> > ### Author Response · Authors · 2023-11-23
> > **Follow-up responses to reviewer Wq6M: Part 2**
> >
> > **4. "Through this way, amino-acid combinations frequently occur in the same context would draw higher attention scores". Have you checked this statement? Could you provide some visualizations of the attention map and correlated frequences of amino-acid combinations?**
> >
> > Ans: Thank the reviewer for this suggestion. We randomly pick one sentence in the training set and visualization its attention matrix from the last layer in Appendix Figure 7(a) and also visualize the pairwise amino acid co-occurrence matrix in Figure 7(b). We find it’s consistent with our statement that residues tend to more connect to their neighboring residues. However, we don’t see any strong correlation between the attention matrix and the pairwise amino acid co-occurrence. Therefore, we revised the corresponding statement in our paper and uploaded a revised version accordingly.
> >
> > **5. Updating residue representations and coordinates in 3D space with only nearest neighbors enables more efficient and economic message passing compared to prior approaches which compute messages on the complete pairwise residue graph". To my knowledge, most of the previous methods such as proteinmpnn adopt knn for constructing sparse graph in the 3D spcace. You should provide evidence to support the significance of your statement.**
> >
> > Ans: We mean from the architecture level, previous methods like SE(3)-Transformer [1] and EGNN [2] adopts information flow from all other atoms.
> >
> > [1] SE(3)-Transformers: 3D Roto-Translation Equivariant Attention Networks. Fabian B. Fuchs et al. NeurIPS 2020.
> >
> > [2] E(n) Equivariant Graph Neural Networks. Victor Garcia Satorras et al. ICML 2021.
> >
> > **6. Novelty: I do not observe enough novelty from the perspective of machine learning in algorithm design.**
> >
> > Ans: Yes, from machine learning algorithm level, our method is not new. However, as admitted by reviewer MgyU, we design a new architecture for protein design. The key innovations of our architecture include: (1) interleaving layers of sequence-level attention and local neighborhood equivariant sub-layer (2) accelerating the local sub-layer with the k-nearest neighbors.

---

> > > ### Comment · Reviewer_Wq6M · 2023-11-23
> > > **To authors**
> > >
> > > The original manuscript of this article has too many issues, despite the author's revisions based on the reviewers' feedback, which also raised doubts about the quality of the original submission. After all, the time is for reviewers to conduct their review and there may be more undiscovered problems. In addition, many identified issues still remain unresolved:
> > > - The paper emphasizes sequence-structure consistency as a key motivation. However, the experimental results do not support this claim, as RFDiffusion+IF achieves a better consistency score.
> > > - It is unclear whether the proposed method can extend to and achieve good performance on standard inpainting datasets, as the authors have not provided corresponding results.
> > > - The proposed method demonstrates superior performance compared to baselines on two protein structures, as it is specifically trained on such data. In contrast, the baseline method is not intended to be trained on two proteins. Therefore, the comparison itself may be unfair.
> > > - The novelty of the work is limited, which the authors acknowledge.
> > >
> > > Considering the aforementioned reasons, I maintain my initial score. I encourage the authors to revise the manuscript, eliminate statements that cannot be supported by results, and perform comparisons on standard datasets. If the authors address these concerns and make the necessary revisions, and if I serve as the reviewer in the next round, I will support the acceptance of the paper. At this stage, I believe the paper falls below the acceptance criteria of ICLR.

---

### Official Review · Reviewer_VH6U · 2023-11-01

**Soundness:** 2 fair
**Presentation:** 3 good
**Contribution:** 3 good
**Rating:** 3
**Confidence:** 4

**Summary:**

This paper proposed a new model called NAEPro to update the coordinate and AA types of proteins. In comparison to other baseline methods on the two protein datasets, NAEPro achieves the top recovery rate, TM-score, and the lowest RMSD. In silico evaluations also confirmed the model's ability to design proteins for binding to their target metallocofactors.

**Strengths:**

The paper is easy to read.

**Weaknesses:**

1. The identified problem or the motivation for designing such a method is not supported by existing literature (e.g., Questions 1-2);
2. Some designed modules counter the intuition in biology (e.g., Questions 5);
3. The method is not 'extensively evaluated' as claimed by the authors (they only selectively analyzed 2 proteins), and many results do not suggest that the proposed method is SOTA among baselines. Moreover, the evaluation does not include all relevant methods, which makes it too aggressive for the authors to claim in the Introduction that their method "achieves the HIGHEST...among ALL competitors" and "is faster than the FASTEST method".
4. Results in Table 1 do not support the superiority of the proposed method. For instance, RMSD>1.7 \AA means the two structures are different for myoglobin. No matter if the predicted result is 4 \AA away for 10 \AA away, neither of them are reliable prediction. For pLDDT, the prediction on $\beta$-lactamase is ~66%, which is much lower than 80%-90% which is conventionally believed reliable for folding predictions. The TM-score, if they are smaller than 0.5, then the two structures are believed different. In this case, both 0.48 and 0.46 are considered a failed design, and it is meaningless to yield the 2% outperformance in this case.

**Questions:**

1. Page 1: "The use of separate models for sequence and structure cannot ensure the consistency between the generated sequence and structure." I disagree with this statement. Generated protein sequences with higher than 30% identity to the wild-type protein merely change their geometry in space, while existing de novo methods usually generate new sequences with 30-90% sequence identity. Empirically, It has been proven by several recent research that the de novo method can generate new sequences for fixed backbone or desired functions [1-3].
2. Page 2: "knowing the topology of a protein before design process is difficult and also cannot guarantee the designed proteins have the desired functions." I disagree with this statement. Although AlphaFold2 cannot predict structures for all proteins with no error, it's a great reference, and designing and screening with it is sufficiently powerful in many research work, including those conducted wet-lab experiments.
3. Page 2: what is the meaning of "generally-encoded" for 20 types of amino acids?
4. Page 3: "The selection of motif varies from setting to setting..." I cannot understand this example regarding the difference between motifs for active sites and binding sites. Can the authors please provide a clearer explanation?
5. Page 5: "motif mining". I don't see the rationale of this module. MSA essentially discovers the conserved sites of a protein sequence by comparing them with other sequences from the same family. However, these conserved sites might not be directly related to its central function. For instance, many conserved sites are buried within the proteins, but functional sites are generally exposed at the surface of the protein. Moreover, the active sites are usually spatially close, but they may distance sequentially.
6. Page 7: "evaluation metrics": ESMFold, as observed by their authors, performs as well as AlphaFold2 when the folded sequence exists in nature. However, for a novel sequence, its prediction is worse than AlphaFold2. Consequently, evaluating novel sequences should use AlphaFold2 instead of ESMFold.
7. Page 7: "Consistency - the RMSD between the designed structure and predicted one". I cannot understand this statement. What is the designed structure and the predicted structure? Which is from NAEPro? Where does the other one come from?



Reference:
1. Watson et al., De novo design of protein structure and function with RFdiffusion (2023).
2. Sumida et al., Improving protein expression, stability, and function with ProteinMPNN (2023).
3. Zhou et al., Conditional Protein Denoising Diffusion Generates Programmable Endonuclease Sequences (2023).

---

> ### Author Response · Authors · 2023-11-17
> **Responses to reviewer VH6U**
>
> We thank the reviewer for the insightful questions. We have clarified all your concerns, added the experiments and updated the paper accordingly. Answers to specific points are provided below:
>
> **Weakness 1 & weakness 2 & Q5: The identified problem or the motivation for designing such a method is not supported by existing literature.  Some designed modules counter the intuition in biology, such as "motif mining"**
>
> Ans: We follow the design insight proposed by [1], which starts from the functional site and fills in additional sequence and structure to create a viable protein scaffold in a single forward pass. Different from [1], we provide the functional sites which are automatically mined from MSA as previous methods [2, 3] while [1] provides the functional sites manually. Besides, we design a new architecture called NAEPro to generate the whole protein sequence and structure based on the given partial protein fragments, while [1] achieves this goal by finetuning the RoseTTAFold. However, we agree with the reviewer’s opinion in question 5 that MSA discovers not only protein functional sites as addressed in [2,3] but also finds conversed sites which may not directly relate to the protein function. Therefore, we change all the term “motif/functional sites” in our paper to  “meaningful protein fragments”.
>
> [1] Scaffolding protein functional sites using deep learning.  Jue wang. et al. Science. 2022.
>
> [2] Emerging methods in protein co-evolution. David de Juan et al. Nature Review Genetics. 2013.
>
> [3] Evolutionary information for specifying a protein fold. Michael Socolich et al. Nature. 2005.
>
> **Weakness 3 & Q1: The method is not 'extensively evaluated' as claimed by the authors (they only selectively analyzed 2 proteins), and many results do not suggest that the proposed method is SOTA among baselines. Moreover, the evaluation does not include all relevant methods, which makes it too aggressive for the authors to claim in the Introduction that their method "achieves the HIGHEST...among ALL competitors" and "is faster than the FASTEST method". Empirically, It has been proven by several recent research that the de novo method can generate new sequences for fixed backbone or desired functions [4-6].**
>
> Ans: Sorry for the confusion. Our method achieves the best performance on 4/5 metrics on myoglobin and 3/5 metrics on beta-lactamase among all the compared representative baselines in our paper. Our model is faster than all baselines compared in our paper. We have revised all these statements in our paper (highlighted in red). For other relevant methods as the reviewer suggested in reference [4-6], we have already compared with [4]. [5] was posted on Oct. 3, 2023, which is later than the ICLR submission ddl, and [3] was posted on Aug. 14, 2023, which is within 3 months before ICLR submission ddl. We appreciate the reviewer’s suggestion, and we tried to compare with [3], but they haven’t released their code now. We emailed the authors of [3] to ask for code and they said their work is under review now and refused to provide code currently. We are trying to reimplement the model now and will update the results in our revised version.
>
> [4] Watson et al., De novo design of protein structure and function with RFdiffusion (2023).
>
> [5] Sumida et al., Improving protein expression, stability, and function with ProteinMPNN (2023).
>
> [6] Zhou et al., Conditional Protein Denoising Diffusion Generates Programmable Endonuclease Sequences (2023).

---

> > ### Author Response · Authors · 2023-11-17
> > **Responses to reviewer VH6U: Part 2**
> >
> > **Weakness 4:  Results in Table 1 do not support the superiority of the proposed method. For instance, RMSD>1.7 \AA means the two structures are different for myoglobin. For pLDDT, the prediction on lactamase is ~66%, which is much lower than 80%-90% which is conventionally believed reliable for folding predictions. The TM-score, if they are smaller than 0.5, then the two structures are believed different.**
> >
> > Ans: We would like to clarify our goal more clearly.  Our task is to design the sequence and structure of proteins with effective functions.  We calculated TM-score by AlphaFold2 as suggested by the reviewer, and the two proteins designed by our model both achieves an average score over 0.5 (myoglobin **0.5912** and beta-lactamase **0.5009**). However, as found by previous method [7],  protein pairs with a TM-score >0.5 are most likely in the same fold while those with a TM-score <0.5 are mainly not in the same fold. Therefore, low TM-score can not indicate the designed protein doesn’t have the corresponding function. As claimed in previous work [9,10], pLDDT between 70~90 are classified to be confident. We calculated pLDDT by AlphaFold2 as suggested by the reviewer and the two protein families designed by our model are confident on an average aspect (myoglobin **91.4139** and beta-lactamase **76.9550**).  Similarly, as observed in previous work[11,12], pLDDT has a weak correlation with protein function. Therefore, a moderate TM-score and pLDDT can not indicate a protein expresses no or poor function. As shown in figure 5(b), the designed myoglobin has a RMSD of 3.943\AA with the most similar one in PDB, but it can still bind the corresponding metallocofactor heme in the AlphaFill simulation [8]. To systematically illustrate the function of our designed proteins, we use docking method Gnina [13] to compute the binding affinity between the designed metalloproteins and the corresponding metallocofactors. The results in the following table shows that our model achieves the best binding affinity scores on both metalloproteins, demonstrating the proteins designed by our model are highly potential to actively exhibit biochemical functions.
> >
> > |Model|myoglobin (kcal/mol)|beta-lactamase (kcal/mol)|
> > |:-----|:----:|:----:|
> > |Hallucination|-7.23|-6.37|
> > |Inpainting|-9.32|-7.63|
> > |SMCDiff+ProteinMPNN|-8.79|-6.89|
> > |PROTSEED|-10.07|-7.68|
> > |FrameDiff+ProteinMPNN|-9.54|-7.21|
> > |RFDiffusion+ProteinMPNN|-9.76|-7.59|
> > |NAEPro|**-10.14**|**-7.80**|
> >
> > [7] How significant is a protein structure similarity with TM-score= 0.5? Jinrui Xu and Yang Zhang. *Bioinformatics. 2010.*
> >
> > [8] Alphafill: enriching alphafold models with ligands and cofactors. Maarten L Hekkelman et al. Nature methods. 2023.
> >
> > [9] pLDDT Values in AlphaFold2 Protein Models Are Unrelated to Globular Protein Local Flexibility. Oliviero Carugo. Crystals. 2023
> >
> > [10] Generating new protein sequences by using dense network and attention mechanism. Feng Wang et al. Mathematical Biosciences and Engineering. 2023.
> >
> > [11] Using AlphaFold to predict the impact of single mutations on protein stability and function. Marina A. Pak et al. PloS one. 2023.
> >
> > [12] Peptide binder design with inverse folding and protein structure prediction. Patrick Bryant et al. Nature Communications Chemistry. 2023.
> >
> > [13] GNINA 1.0: Molecular docking with deep learning. A McNutt et al. Cheminformatics. 2021.
> >
> > **Q1 & Q2: "The use of separate models for sequence and structure cannot ensure the consistency between the generated sequence and structure." "knowing the topology of a protein before design process is difficult and also cannot guarantee the designed proteins have the desired functions." I disagree with this statement.** **I disagree with this statement.**
> >
> > Ans: Thanks for the suggestion! We agree with the reviewer and updated the statement in our paper.
> >
> > **Q3: what is the meaning of "generally-encoded" for 20 types of amino acids?**
> >
> > Ans: It means the 20 common amino acids. We have updated the paper accordingly.
> >
> > **Q4: "The selection of motif varies from setting to setting...". Can the authors please provide a clearer explanation?**
> >
> > Ans: We mean the motifs have different meanings in different tasks. For de novo enzyme design [14], it means the binding sites where enzyme binds the metallocofactor and substrate. For de novo binder design [15], it means the binding sites where the protein binder binds the protein targets.
> >
> > [14] De novo enzyme design using rosetta3. Florian Richter et al. PloS one. 2011.
> >
> > [15] De novo design of protein interactions with learned surface fingerprints. Pablo Gainza. Nature. 2023.

---

> > > ### Author Response · Authors · 2023-11-17
> > > **Responses to reviewer VH6U: Part 3**
> > >
> > > **Q6: For a novel sequence, ESMFold’s prediction is worse than AlphaFold2. Consequently, evaluating novel sequences should use AlphaFold2 instead of ESMFold.**
> > >
> > > Ans: Due to limited time and computing resources, we provide the AlphaFold2 results for ProtSeed, RFDiffusion+ProteinMPNN and our NAEPro. As the results show, even though ESMFold achieves lower scores than AlphaFold2, the tendencies (ranking) among different models are similar.  Particularly, our model achieves an average TM-score over 0.5 on beta-lactamase (ESMFold 0.4812) and pLDDT over 70 (ESMFold 66.8705), which gives stronger evidence that our model has the ability to design proteins which have stable structures.
> > >
> > > **myoglobin**
> > >
> > > |Model| pLDDT|TM-score|
> > > |:-----|:----:|:----:|
> > > |PROTSEED|77.9032|0.5621|
> > > |RFDiffusion+ProteinMPNN|88.2130|0.5103|
> > > |NAEPro|**91.4139**|**0.5912**|
> > >
> > > **$\beta$-lactamase**
> > >
> > > |Model| pLDDT|TM-score|
> > > |:-----|:----:|:----:|
> > > |PROTSEED|73.3176|0.4901|
> > > |RFDiffusion+ProteinMPNN|**87.8092**|0.4806|
> > > |NAEPro|76.9550|**0.5009**|
> > >
> > >  **Q7: "Consistency - What is the designed structure and the predicted structure? Which is from NAEPro? Where does the other one come from?**
> > >
> > > Ans: Consistency is calculated as the RMSD between the designed structure from NAEPro and the one predicted by ESMFold for the designed sequence from NAEPro.

---

> > > > ### Author Response · Authors · 2023-11-21
> > > > **Looking forward to any other questions!**
> > > >
> > > > Thanks for your valuable review and comments again. We have addressed all your concerns and questions.
> > > >
> > > > If you have any other questions, please feel free to discuss with us.

---

> ### Comment · Reviewer_VH6U · 2023-11-22
>
> Thanks for your responses. However, I remain unconvinced regarding the issues I raised earlier.
>
> **Response to Weakness 3 & Q1**: My point in mentioning these three papers is not to introduce additional (and possibly meaningless) in-silico comparisons to the authors. Instead, they validated their designs with solid wet-lab experiments demonstrating that it is possible to generate novel protein sequences with **fixed** backbones.
>
> **Response to Weakness 4**: I completely agree that all the existing evaluation metrics we used in deep learning are only indirect indicators, and they only imply a higher chance that the assessed protein will perform its function. However, I have trouble following the logic that "*Therefore, a low TM-score cannot indicate the designed protein doesn’t have the corresponding function*" suggests "proteins with TM-scores that are lower than 0.5 (which is a wildly-used standard) are still reliable in terms of their functionality in general." Similarly, if we all agree that the existing metrics are suboptimal and none of them provides 100% accurate evaluations, it is then meaningless to criticize the widely applied standards with one or two counter-examples. After all, you don't have any better evidence (such as wet-lab results) that is more reliable than the existing criteria.
>
> **Response to Q6**: Because ESMFold is a less reliable tool for evaluating novel proteins (as mentioned in their paper), it should not be used here to assess your designs at all -- If the evaluation metric itself is unreliable, the whole evaluation makes no sense.

---

> > ### Author Response · Authors · 2023-11-23
> > **Follow-up rebuttal to reviewer VH6U**
> >
> > Thanks for the further response! Please see our responses to your follow-up questions and concerns as follows:
> >
> > **Response to Weakness 3 & Q1: My point in mentioning these three papers is not to introduce additional (and possibly meaningless) in-silico comparisons to the authors. Instead, they validated their designs with solid wet-lab experiments demonstrating that it is possible to generate novel protein sequences with fixed backbones.**
> >
> > Ans: We agree on the reviewer’s opinion and have accordingly updated the statement in Introduction.  We have uploaded a revised version of our paper.
> >
> > **Response to Weakness 4: I completely agree that all the existing evaluation metrics we used in deep learning are only indirect indicators, and they only imply a higher chance that the assessed protein will perform its function. However, I have trouble following the logic that "*Therefore, a low TM-score cannot indicate the designed protein doesn’t have the corresponding function*" suggests "proteins with TM-scores that are lower than 0.5 (which is a wildly-used standard) are still reliable in terms of their functionality in general." Similarly, if we all agree that the existing metrics are suboptimal and none of them provides 100% accurate evaluations, it is then meaningless to criticize the widely applied standards with one or two counter-examples. After all, you don't have any better evidence (such as wet-lab results) that is more reliable than the existing criteria.**
> >
> > Ans: We agree with Reviewer VH6U. As suggested by reviewer MgyU, the binding affinity is a better metric for functional protein design. We calculated the binding affinity scores for top-5, top-10, and top-30 candidates and also median score on the test set using Gnina. The results are reported as follows:
> >
> > **Binding affinity score for designed myoglobin**
> >
> > |Model| top-5 (kcal/mol)|top-10 (kcal/mol)|top-30 (kcal/mol)| median|
> > |:-----|:----:|:----:|:----:|:----:|
> > |Hallucination|-8.18$\pm$ 0.01|-8.07$\pm$0.03|-7.97$\pm$0.23|-7.25|
> > |Inpainting|-13.47$\pm$0.02|-13.12$\pm$0.12|-12.31$\pm$0.54|-9.56|
> > |SMCDiff+ProteinMPNN|-11.37$\pm$0.03|-11.12$\pm$0.31|-10.87$\pm$0.42|-8.76|
> > |PROTSEED|-13.21$\pm$0.13|-12.89$\pm$0.42|-11.98$\pm$0.52|-10.23|
> > |FrameDiff+ProteinMPNN|-13.13$\pm$0.05|-12.92$\pm$0.16|-12.21$\pm$0.23|-10.08|
> > |RFDiffusion+ProteinMPNN|-13.68$\pm$0.02|-13.03$\pm$0.21|-12.56$\pm$0.43|-10.15|
> > |NAEPro|**-14.12$\pm$0.01**|**-13.85$\pm$0.10**|**-13.06$\pm$0.38**|**-10.74**|
> >
> > **Binding affinity score for designed beta-lactamase**
> >
> > |Model| top-5 (kcal/mol)|top-10 (kcal/mol)|top-30 (kcal/mol)| median|
> > |:-----|:----:|:----:|:----:|:----:|
> > |Hallucination|-6.98$\pm$0.01|-6.87$\pm$0.02|-6.69$\pm$0.05|-6.29|
> > |Inpainting|-9.89$\pm$0.03|-9.54$\pm$0.16|-9.13$\pm$0.43|-7.24|
> > |SMCDiff+ProteinMPNN|-9.10$\pm$0.01|-9.05 $\pm$0.02|-8.98$\pm$0.01|-6.97|
> > |PROTSEED|-9.88$\pm$0.21|-9.51$\pm$0.41|-9.01$\pm$0.62|-7.31|
> > |FrameDiff+ProteinMPNN|-9.54$\pm$0.03|-9.56$\pm$0.23|-8.89$\pm$0.35|-7.03|
> > |RFDiffusion+ProteinMPNN|-9.87$\pm$0.05|-9.56$\pm$0.23|-9.12$\pm$0.53|-7.51|
> > |NAEPro|**-10.06$\pm$0.05**|**-9.79$\pm$0.10**|**-9.39$\pm$0.12**|**-7.66**|
> >
> > **Response to Q6: Because ESMFold is a less reliable tool for evaluating novel proteins, it should not be used here to assess your designs at all.**
> >
> > Ans: We totally agree with reviewer VH6U. We provide the mean pLDDT for top-5, top-10, top-30 candidates calculated by AlphaFold2 as follows:
> >
> > **AlphaFold2 pLDDT for designed myoglobin**
> >
> > |Model| top-5|top-10|top-30|
> > |:-----|:----:|:----:|:----:|
> > |Hallucination|76.7243|73.5953|68.6060|
> > |Inpainting|97.6171|97.6063|97.5681|
> > |SMCDiff+ProteinMPNN|85.7051|84.7744|84.9220|
> > |PROTSEED|92.9770|91.9216|90.4293|
> > |FrameDiff+ProteinMPNN|97.3997|97.3329|96.4279|
> > |RFDiffusion+ProteinMPNN|97.9722|97.9174|97.6063|
> > |NAEPro|**98.1057**|**98.0706**|**97.9197**|
> >
> > **AlphaFold2 pLDDT for designed $\beta$-lactamase**
> >
> > |Model| top-5|top-10|top-30|
> > |:-----|:----:|:----:|:----:|
> > |Hallucination|74.7760|74.7032|71.2932|
> > |Inpainting|94.5489|94.4328|93.2941|
> > |SMCDiff+ProteinMPNN|86.2510|85.7072|83.0376|
> > |PROTSEED|98.2896|98.2740|97.9683|
> > |FrameDiff+ProteinMPNN|98.3049|98.2922|98.1267|
> > |RFDiffusion+ProteinMPNN|**98.6225**|**98.5116**|98.0643|
> > |NAEPro|98.5385|98.4481|**98.2841**|

---

> > > ### Author Response · Authors · 2023-11-23
> > > **Looking forward to any further discussion!**
> > >
> > > We much appreciate your valuable comments and insightful suggestions again, which helps a lot to improve the quality of our paper. As the discussion deadline is approaching, please let us know if you have any further concerns or questions. We are happy to have any further discussion!

---

### Author Response · Authors · 2023-11-17
**Summary of Revisions**

We thank all the reviewers' valuable suggestions.

We have uploaded a revised draft that incorporates feedback from reviewers VH6U and MgyU, with the updated portions highlighted in red. Additionally, in response to reviewer Wq6M's suggestion, we have included the code and data we used in our paper in the supplementary material (Due to the maximum memory limitation -100M, we couldn’t upload the model checkpoints).

Here's an overview highlighting the principal modifications to our paper:

1. The revised term from “motif/functional sites” to “meaningful fragments” in abstract, introduction third paragraph, method opening paragraph, method 3.1-3.4, experiments 4.1-4.2, analysis 5.1, Conclusion (Reviewer VH6U, weakness 1 & weakness 2 & Q5)
2. The corrected claim in abstract and last paragraph in introduction (review VH6U, weakness3)
3. The corrected claim in Introduction second paragraph (Reviewer VH6U, Q1)
4. The corrected claim in Related Work second paragraph (Reviewer VH6U, Q2)
5. The clarified explanation of 20 common amino acids in Method opening paragraph(Reviewer VH6U, Q3)
6. The corrected theoretical claim in Method 3.5 and Appendix A. 1 (reviewer MgyU, weakness 5)
7. The clarified task definition in Method opening paragraph (reviewer MgyU, Q2)

We look forward to any additional reading and feedback. If you have any further questions, we are very happy to continue discussion!

---

### Meta-Review · Area_Chair_9pPe · 2023-12-05

**Metareview:**

NAEPro, a new model introduced in the paper, is aimed at updating protein coordinates and amino acid types, with a focus on jointly designing protein sequences and structures. The model utilizes a combination of attention and equivariant layers, designed to capture both global and local patterns in protein sequences and their 3D structures. It incorporates aspects of ESM-2 and EGNN, resembling transformer architectures used in protein motif-scaffolding problems. The paper also presents new datasets, including beta-lactamase and myoglobin, for evaluating scaffolding performance. In comparative studies, NAEPro shows interesting results, with some improvements in recovery rates, TM-scores, and RMSD, suggesting its potential utility in protein design and motif scaffolding applications.

**Justification For Why Not Higher Score:**

Reviewers argue that the paper, while presenting innovative aspects, faces several key weaknesses. Firstly, the claimed sequence-structure consistency of NAEPro is not convincingly demonstrated, as another model, RFDiffusion+IF, shows better consistency scores. Additionally, the paper does not address NAEPro's performance on standard datasets, leaving its broader applicability unclear. The model's comparative analysis also raises fairness concerns, as it outperforms baselines on data specifically tailored for NAEPro, unlike the more general-purpose baseline models. Finally, the authors themselves acknowledge the limited novelty of their work, suggesting that while NAEPro introduces certain advancements, its overall innovation might be constrained.

**Justification For Why Not Lower Score:**

N/A

---

### Decision · Program_Chairs · 2024-01-16

Reject